# Opponent regulation of action performance and timing by striatonigral and striatopallidal pathways

Konstantin I Bakhurin[1†], Xiaoran Li[1†], Alexander D Friedman[1], Nicholas A Lusk[1], Glenn DR Watson[1], Namsoo Kim[1], Henry H Yin[1,2*]

[1]Department of Psychology and Neuroscience, Duke University, Durham, United States; [2]Department of Neurobiology, Duke University School of Medicine, Durham, United States

**Abstract** The basal ganglia have been implicated in action selection and timing, but the relative contributions of the striatonigral (direct) and striatopallidal (indirect) pathways to these functions remain unclear. We investigated the effects of optogenetic stimulation of D1+ (direct) and A2A+ (indirect) neurons in the ventrolateral striatum in head-fixed mice on a fixed time reinforcement schedule. Direct pathway stimulation initiates licking, whereas indirect pathway stimulation suppresses licking and results in rebound licking after stimulation. Moreover, direct and indirect pathways also play distinct roles in timing. Direct pathway stimulation produced a resetting of the internal timing process, whereas indirect pathway stimulation transiently paused timing, and proportionally delayed the next bout of licking. Our results provide evidence for the continuous and opposing contributions of the direct and indirect pathways in the production and timing of reward-guided behavior.

**\*For correspondence:**
hy43@duke.edu

†These authors contributed equally to this work

**Competing interests:** The authors declare that no competing interests exist.

## Introduction

The striatum is the major input nucleus of the basal ganglia (BG) and contains two major populations of medium spiny projection neurons (*Gerfen et al., 1990*; *Mink, 1996*). Striatonigral (direct pathway) neurons express D1 receptors and project to the substantia nigra pars reticulata (SNr) and other BG output nuclei. Striatopallidal (indirect pathway) neurons express D2 receptors and project to the external globus pallidus, which in turn inhibits the SNr. Ever since the connectivity of these pathways was defined, there has been controversy on their contributions to behavior. Given their opposite effects on BG output, an influential model of the striatum casts the direct and indirect pathways in opposing roles with the direct pathway promoting behavior and the indirect pathway suppressing behavior (*Albin et al., 1989*; *Freeze et al., 2013*; *Kravitz et al., 2010*). More recently, however, this view has been questioned by studies that described concurrent activation of these two populations during behavior, suggesting complementary contributions of these two pathways (*Cui et al., 2013*; *Isomura et al., 2013*).

The lateral or sensorimotor striatum receives topographically organized innervation from the cortex and thalamus (*Hintiryan et al., 2016*). Micro-stimulation of different striatal regions results in movements of different body parts (*Alexander and DeLong, 1985*). In rats, the ventrolateral striatum (VLS) has been implicated in orofacial behaviors, including licking (*Mittler et al., 1994*; *Pisa, 1988a*; *Pisa, 1988b*). Although licking behavior is generated by brain stem pattern generators (*Travers et al., 1997*), BG output can provide top-down regulation of the pattern generators, either directly or via the nigrotectal projections (*Deniau and Chevalier, 1992*; *Redgrave et al., 1992*; *Rossi et al., 2016*; *Shammah-Lagnado et al., 1992*; *Toda et al., 2017*). Stimulation or enhancing dopaminergic signaling in the VLS can generate orofacial behavior, in some cases dyskinetic

movements, while dopamine depletion can abolish such behaviors (*Delfs and Kelley, 1990*; *Jicha and Salamone, 1991*; *Kelley et al., 1988*). However, the detailed pathway-specific mechanisms underlying such observations remain largely unknown.

One important characteristic of self-initiated behaviors is that they can be precisely timed relative to some anticipated event like a reward (*Gibbon et al., 1984*; *Pavlov, 1927*; *Roberts, 1981*). The BG have been implicated in such interval timing in the seconds to minutes range (*Buhusi and Meck, 2005*; *Merchant et al., 2013*). Manipulation of dopamine activity can bidirectionally influence estimation of the passage of time (*Buhusi and Meck, 2005*; *Meck, 1996*) and lesions of the striatum also impair timing behavior (*Meck, 2006*). Furthermore, striatal activity has also been shown to represent time intervals (*Bakhurin et al., 2017*; *Gouvêa et al., 2015*). Although the role of D1-expressing and D2-expressing neurons in timing has been studied using systemic pharmacological manipulations (*Frederick and Allen, 1996*), it remains unclear how the two pathways of the BG interact to mediate these behaviors.

In this study, we used optogenetic stimulation to examine the contributions of the VLS in both action generation and timing. Using a fixed-time (FT) schedule of reinforcement, combined with occasional peak probe trials, we showed that mice rapidly acquired highly predictable anticipatory licking patterns that reflects reliable internal timing processes (*Toda et al., 2017*). Using temporally and spatially precise manipulations of neural activity with optogenetics, we studied how direct and indirect pathways in the VLS regulate the initiation, maintenance, and timing of self-initiated licking.

## Results

To study the roles of the direct and indirect pathways during self-initiated behavior, we manipulated the activity of cell populations that originate these pathways in the orofacial region of the striatum (*Figure 1A & B*). We bilaterally injected a Cre-dependent AAV5-DIO-ChR2 virus in the VLS region of the striatum in D1-Cre (*Figure 1C*) and A2A-Cre (*Figure 1D*) mice and implanted chronic optic fibers just above this area. Five additional A2A-Cre mice received AAV5-DIO-eYFP virus along with fiber implants. We tracked licking behavior in head-fixed mice performing an interval timing task (*Toda et al., 2017*). Mice received a 5 µL drop of 10% sucrose solution every 10 s (*Figure 1E*). Voluntary licking behavior in all mice consisted of bouts of rhythmic protrusion and retractions of the tongue at a relatively fixed rate (4–8 Hz). Well-trained mice reliably generated anticipatory licking behavior before reward delivery (*Figure 1F*). As a result, the average licking rate gradually ramps up in expectation of reward, and then peaks following reward and shows a sharper decline after consumption (*Figure 1G*).

We tested the effects of stimulating the direct pathway in the VLS on licking at three different time points during the interval. We stimulated the direct pathway for 1 s before mice normally initiate anticipatory licking, starting at 5 s before reward delivery (*Figure 2A*). Though licking was sparse at this time point (*Figure 2B*, top), activation of direct pathway MSNs induced licking (*Figure 2B*, bottom and *Figure 2C*, *Video 1*). We determined the change in lick rate that resulted from laser stimulation by calculating the lick rate during the 1 s laser stimulation period for D1-Cre mice and a control group that received eYFP virus. The normal lick rate obtained for the same time point during trials without laser stimulation was subtracted from this value to quantify the stimulation-induced change in lick rate (*Figure 2D*). A two-way ANOVA (stimulation frequency x experimental group) revealed a change of licking rate resulting from VLS direct pathway stimulation (main effect of group, $F_{1,44} = 93.6$, p<0.0001) and a significant effect of laser frequency ($F_{3,44} = 4.64$, p<0.01); there was a also an interaction between frequency and group ($F_{3,44} = 4.84$, p<0.01). Post-hoc tests revealed that lick rate was significantly increased by 10 Hz (p<0.01), 25 Hz, (p<0.001), and 50 Hz (p<0.001) stimulation of the direct pathway. We next stimulated the direct pathway at 1 s before reward delivery (*Figure 2E*), during the time of maximal anticipatory licking (*Figure 2F*, top). Although there was a ceiling effect on licking at this time point (*Figure 2F*, bottom), we regularly observed a potentiation of the licking rate during stimulation when compared to trials without stimulation (*Figure 2G*). A two-way ANOVA revealed significant increase in licking as a result of VLS stimulation (*Figure 2H*, main effect of group, $F_{1,37} = 48.82$, p<0.0001), but no effect of frequency ($F_{3,37} = 1.31$, p=0.3) nor a significant interaction between frequency and group ($F_{3,37} = 1.79$, p=0.16). Stimulation at the time of reward delivery (*Figure 2I*) had the most modest impact on licking rate (*Figure 2J and K*). However, a two-way ANOVA detected a significant effect of stimulation on licking even during

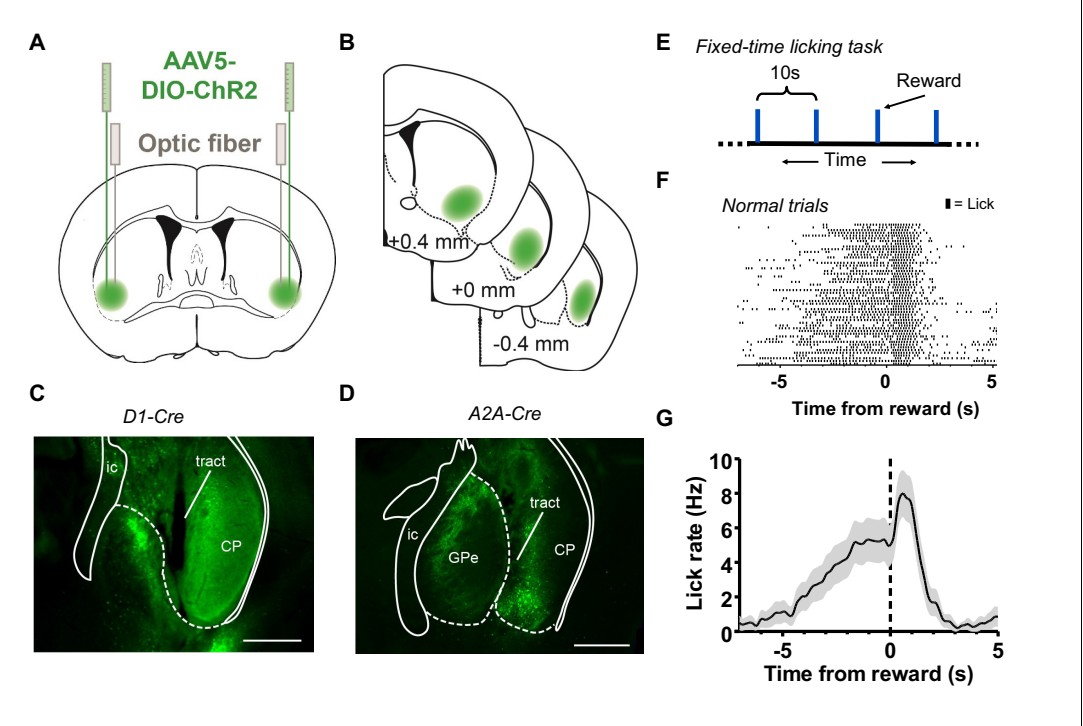

**Figure 1.** Stimulation of the direct and indirect pathways of the orofacial striatum during a licking-based interval timing task. (A) *Left*: Strategy of injection of AAV-ChR2 virus to orofacial area of the striatum in D1-Cre or A2A-Cre mice and implantation of chronic optical fibers over the injection site. (B) Serial section diagrams showing target zone for orofacial striatum. (C) Representative image of eYFP-expression in ventrolateral striatum in a D1-Cre + mouse injected with AAV-DIO-ChR2-eYFP. Scale bar = 500 µm. (D) Representative image of eYFP-expression in an A2A-Cre+ mouse injected with AAV-DIO-ChR2-eYFP virus. Scale bars = 500 µm. (Abbreviations: *ic*, internal capsule; *CP*, caudoputamen; *GPe*, globus pallidus, external segment.) (E) Diagram illustrating the fixed-time interval task for head-fixed mice. Mice receive a 10% sucrose reward (5 µL) every 10 s. (F) Example lick raster of a well-trained mouse during normal trials aligned to reward delivery. Tick marks represent individual licks. (G) Mean licking rate calculated from the raster in *E*. Error bars show SEM.

consumption (Figure 2L, $F_{1,36}$ = 8.65, p<0.01), but did not show a significant effect of stimulation frequency ($F_{3,36}$ = 1.27, p=0.3), nor a significant interaction ($F_{3,36}$ = 1.73, p=0.18). We directly compared the effects of stimulation on licking as a function of time of laser stimulation during the task. A two-way ANOVA revealed significant effects of both the time of stimulation in the task (*Figure 2M*, main effect of time, $F_{3,67}$ = 9.16, p<0.0001) and laser frequency ($F_{2,67}$ = 24.31, p<0.0001), with no significant interaction ($F_{6,67}$ = 0.99, p=0.44). Together, these results show for the first time that direct pathway stimulation in the orofacial striatum can result in the generation of licking behavior. The effects of direct pathway stimulation were dependent on the on-going behavioral state of the animals, which can be explained by a ceiling effect.

Remarkably, we observed that it was possible for licking to exceed the frequencies normally produced by mice during direct pathway stimulation (*Figure 3A*). Licking during laser presentation could be shifted rightward toward a frequency of 10 Hz (*Video 2*). This appeared to be the upper limit of the central pattern generator for licking, as naturally generated licking showed predominant frequencies between 4 and 6 Hz. To explore this effect, we analyzed licking in the frequency domain. We quantified power spectral density (PSD) distributions for licking and observed that licking frequencies are higher during direct pathway stimulation compared to licking produced during anticipation (*Figure 3B*). Laser stimulation increased the licking frequency to around 10 Hz, which is rarely reached by mice naturally. Varying stimulation frequency also systematically regulated the power of licking around a specific frequency, reflected in the percentage of the PSD distribution occupying that frequency range (*Figure 3C*). We compared the amplitude of the PSD in 3 non-overlapping licking frequency bands (4-6, 6-8, and 8-10 Hz) as a function of direct pathway stimulation frequency when stimulation occurred 1 second prior to reward. A two-way mixed ANOVA showed that was a significant interaction between stimulation frequency and licking frequency bands (Figure 3D, $F_{8,58}$ =

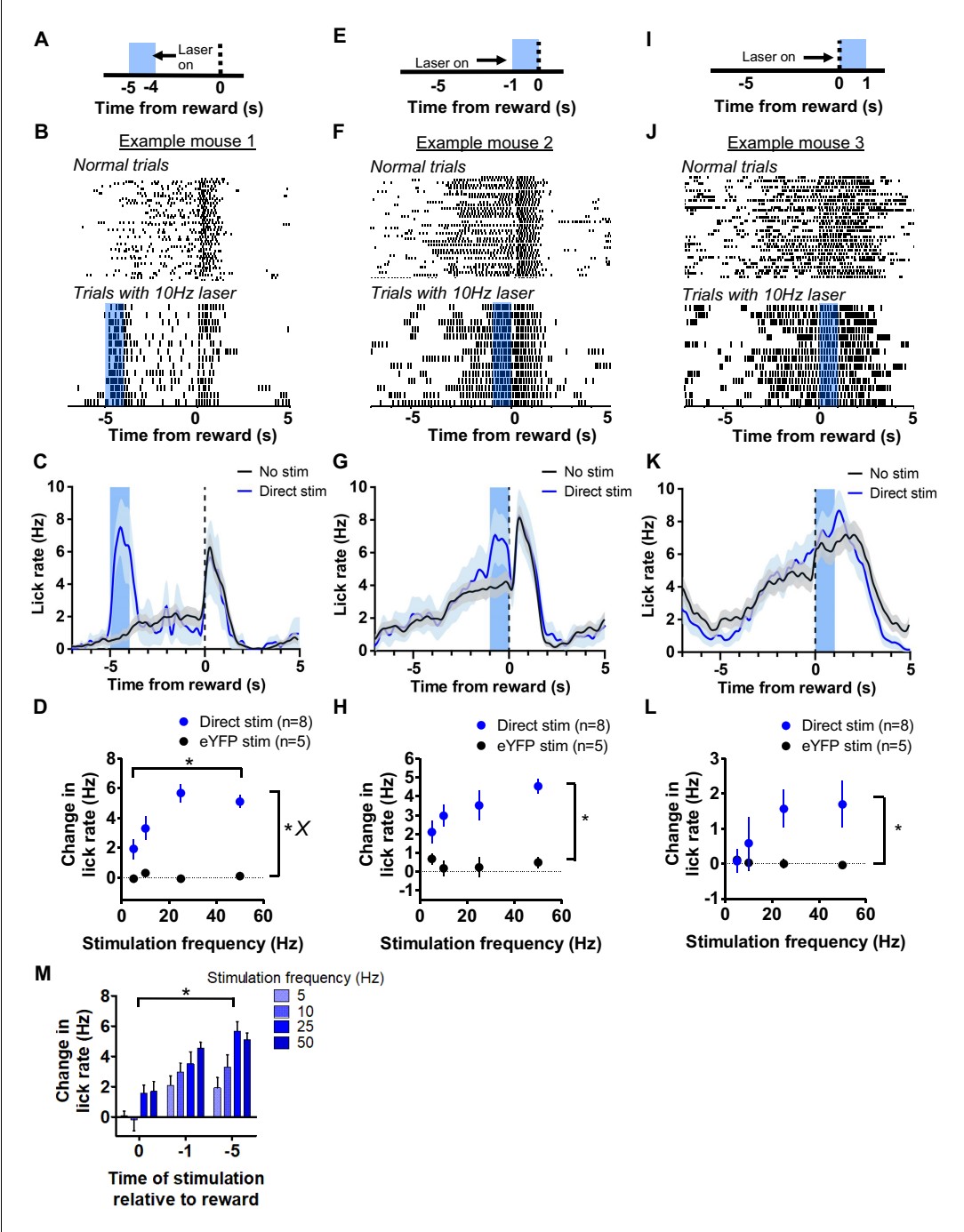

**Figure 2.** Stimulation of direct pathway in VLS generates licking. (**A**) Depiction of stimulation strategy 5 s prior to reward in D1-cre mice. (**B**) *Top*: Peri-event lick raster diagrams of representative trials demonstrating normal anticipatory licking patterns in trials without laser. *Bottom*: 10 Hz stimulation at 5 s prior to reward results in large increases in licking activity. Both rasters reflect licking from the same mouse in the same session. (**C**) Mean lick rate calculated from the example session shown in *B* for trials with and without laser stimulation. (**D**) Mean increases in licking rate during laser delivery resulting from stimulation of D1 MSNs relative to control at 5 s increased as a function of stimulation frequency (n = 8 D1-ChR2 mice, n = 5 control mice; two-way ANOVA; main effect of experimental group, $F_{1,44}$ = 93.6, p<0.0001; main effect of frequency, $F_{3,44}$ = 4.64, p<0.01; interaction between frequency and group, $F_{3,44}$ = 4.84, p<0.01). (**E**) Depiction of stimulation strategy 1 s prior to reward. (**F**) *Top*: Peri-event lick raster diagrams of representative non-laser trials. *Bottom*: Raster diagram of trials containing 10 Hz stimulation during anticipatory licking. Both rasters reflect licking from the same mouse in the same session. (**G**) Mean lick rate calculated from the example session shown in *F* for trials with and without laser stimulation. (**H**) Stimulation of D1 MSNs at 1 s prior to reward increased the rate of anticipatory licking (two-way ANOVA, main effect of group, $F_{1,37}$ = 48.82, p<0.0001). (**I**) Depiction of stimulation strategy coinciding with reward delivery. (**J**) *Top*: Representative lick raster diagrams of licking behavior during trials without

*Figure 2 continued on next page*

*Figure 2 continued*

laser stimulation. *Bottom*: Raster diagram of trials from the same animal containing 10 Hz stimulation during consumption licking. Both rasters reflect behavior recorded in the same session. (K) Mean licking rate calculated from the example session shown in *J* for trials with and without stimulation. (L) Stimulation of D1 MSNs was capable of increasing consumption licking rate (two-way ANOVA, main effect of group, $F_{1,36}$ = 8.65, p<0.01). (M) The time of stimulation during the interval influenced the change in licking rate (two-way ANOVA, main effect of time of stimulation, $F_{3,67}$ = 9.16, p<0.0001; main effect of frequency, $F_{2,67}$ = 24.31, p<0.0001). Error bars show SEM. *X* symbol reflects a significant interaction between factors.

2.54, p < 0.05), and significant main effects of both licking frequency band ($F_{2,58}$ = 15.46, p < 0.0001) and stimulation frequency ($F_{4,58}$ = 3.89, p < 0.05). Post-hoc analyses indicated that power in the 8-10Hz band was significantly greater than power in the 4-6 Hz band during 10 (p < 0.05) and 25 (p < 0.01) Hz stimulation and that power in the 6-8 Hz band was significantly greater than in the 4-6 Hz band during 50 Hz (p < 0.01) stimulation.

When stimulation occurred 5 seconds prior to reward, there was a significant interaction between licking frequency band and stimulation frequency (*Figure 3E*, $F_{8,60}$ = 6.43, p < 0.0001) in their effects on PSD amplitude. We also found main effects of frequency band ($F_{2,60}$ = 37.76, p < 0.0001) and stimulation frequency ($F_{4,60}$ = 2.84, p < 0.05). Post-hoc analyses indicated that power in the 8-10 Hz band was significantly greater than power in the 4-6 Hz band during 10 (p < 0.001), 25 (p < 0.001), and 50 (p < 0.001) Hz stimulation and was greater than power in the 6-8 Hz band during 25 Hz stimulation (p < 0.001). In addition, power in the 6-8 Hz band was significantly greater than power in the 4-6 Hz band during 25 (p < 0.05) and 50 Hz (p < 0.05) Hz stimulation. These results indicate that increasing direct pathway stimulation 5 seconds prior to reward also resulted in more licking in the 8–10 Hz frequency band.

Lastly, when stimulation occurred together with reward, there was a significant interaction between licking frequency band and stimulation frequency (*Figure 3F*, $F_{8,56}$ = 2.59, p < 0.05) in their effects on PSD amplitude. We found main effects of frequency band ($F_{2,56}$ = 54.02, p < 0.0001) and stimulation frequency ($F_{4,56}$ = 3.3, p < 0.05). Post-hoc analyses indicated that power in the 8-10 Hz band was significantly greater than power in the 4-6 Hz band during 5 (p < 0.01, 10 (p < 0.001), 25 (p < 0.001), and 50 (p < 0.001) Hz stimulation and was greater than licking power in the 6-8 Hz band during 25 (p < 0.01) and 50 (p < 0.01) Hz stimulation. In addition, licking power in the 6-8 Hz band was significantly greater than power in the 4-6 Hz band during 5 (p < 0.05) and 10 Hz (p < 0.05) Hz stimulation. These results indicate that increasing direct pathway stimulation during consumption licking also resulted in more licking in at 8 – 10 Hz. Together, light delivery to the VLS resulted in rightward shift in the frequency domain, above the naturally occurring states that mice normally generate. In addition, direct pathway stimulation could increase the degree to which of licking occupied a given frequency band, reflected in the shape of the PSD function. These results suggest a role for the BG in regulating the frequency and duty cycle of the licking oscillator in different modes corresponding to roughly three frequency bands (4-6 Hz, 6-8 Hz, and 8-10 Hz).

When stimulating the direct pathway at 5 s prior to reward, we observed that licking resulting from stimulation only began after a certain delay (*Figure 4A*), and that this delay was reduced with higher stimulation frequencies (*Figure 4B*). A one-way ANOVA revealed a significant effect of stimulation frequency on the latency to initiate licking following the onset of laser stimulation (*Figure 4C*; $F_{3,22}$ = 4.92, p<0.01). The lick bout that resulted from stimulation also lasted longer than the 1 s stimulation period. We counted the number of licks that occurred over the course of one second following the offset of stimulation and compared that

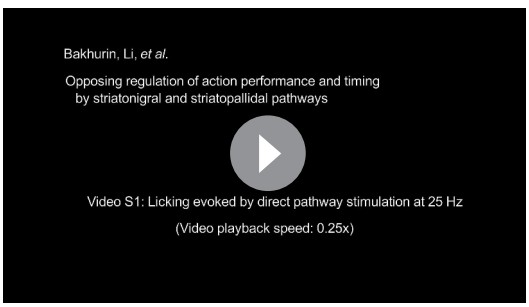

**Video 1.** Licking evoked by direct pathway stimulation at 25 Hz. Video shows a head-fixed D1-Cre mouse with ChR2 expressed in the VLS. The video demonstrates the effect of direct pathway stimulation at 25 Hz 5 s prior to reward delivery, when mice are least likely to lick in the fixed-time task. Stimulation evokes robust licking behavior. Stimulation lasts 1 s, denoted with a blue square in the top right corner. Video playback is at quarter speed.
https://elifesciences.org/articles/54831#video1

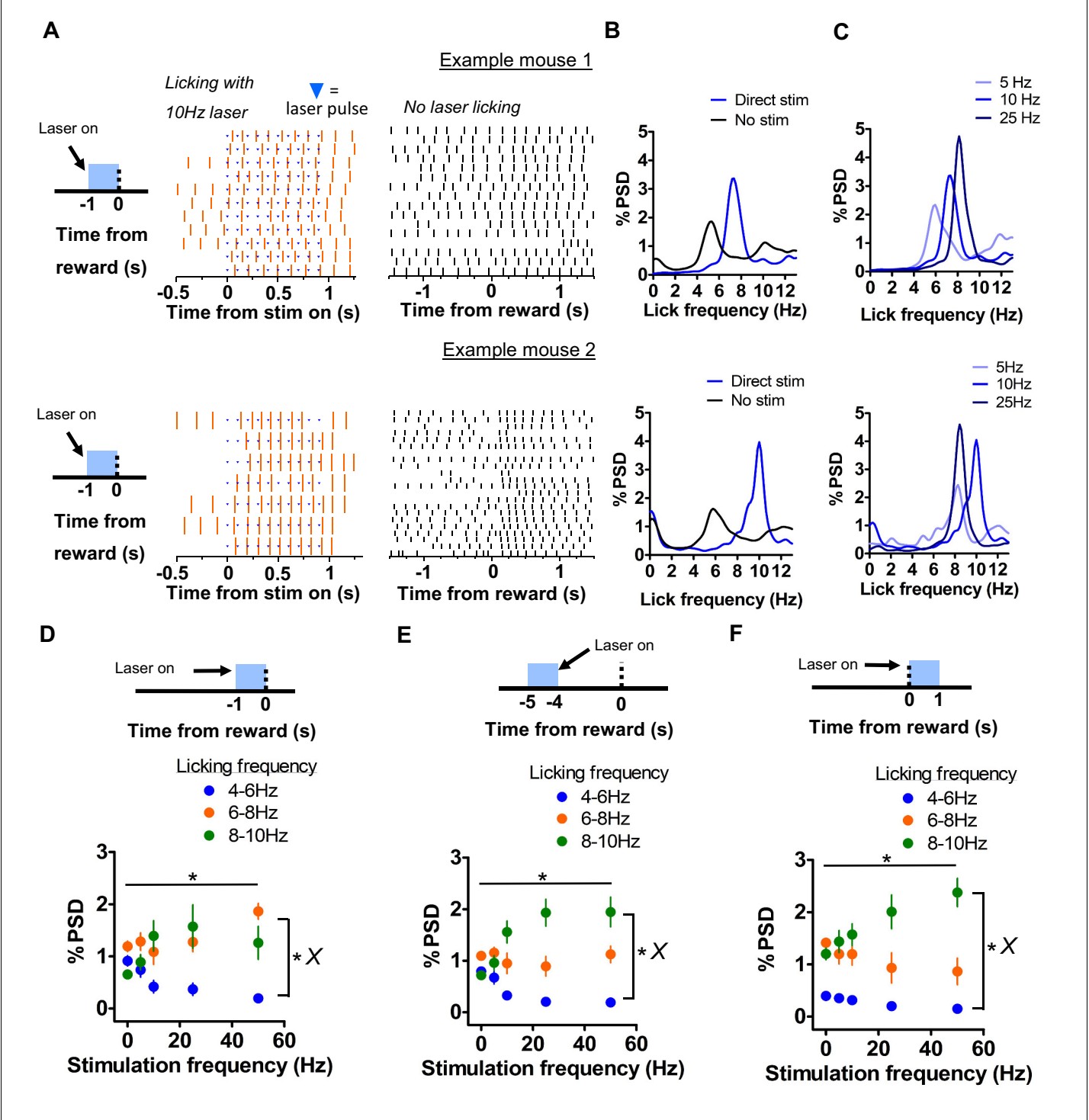

**Figure 3.** Direct pathway boosts licking frequency and modulates licking duty cycle. (A) *Left*: Example lick rasters from two mice showing licking frequency potentiation by direct pathway stimulation 1 second before reward. Note increase of licking to upwards of 10 Hz. Blue tick-marks denote laser pulse times. *Right*: Example lick raster from the same sessions showing normal licking activity around reward. (B) Spectral density analyses of licking during laser stimulation and during anticipation (1 second intervals beginning prior to reward) without stimulation. Analysis of data from the sessions represented in *A* is shown. (C) Power spectral density (PSD) distributions of licking during stimulation at increasing laser frequencies. Note consistent increase in power. (D) Effect of stimulation frequency on occupancy of licking at 3 distinct frequency bands (4-6, 6-8, 8-10 Hz) when stimulation occurred 1 second prior to reward. Peak licking power was related to stimulation frequency (n = 8 mice; two-way mixed ANOVA; interaction band vs stimulation frequency, $F_{8,58} = 2.54$, $p < 0.05$; main effects of frequency band, $F_{2,58} = 15.46$, $p < 0.0001$ and stimulation frequency, $F_{4,58} = 3.89$, p

*Figure 3 continued on next page*

Figure 3 continued

< 0.05). Frequency bands contain non-overlapping ranges, e.g., 4-5.99, 6-7.99, and 8-10. (**E**) Effect of stimulation frequency on occupancy of licking in 3 frequency bands when stimulation occurred 5 seconds before reward. Licking increasingly occupied the 8-10 Hz band with increasing stimulation (two-way mixed ANOVA; interaction band vs frequency, $F_{8,60} = 6.43$, p < 0.0001; main effects of frequency band, $F_{2,60} = 37.76$, p < 0.0001 and stimulation frequency, $F_{4,60} = 2.84$, p < 0.05). (**F**) Effect of stimulation frequency on occupancy of licking in 3 frequency bands when stimulation coincided with reward. Licking increasingly occupied the 8-10 Hz band with increasing stimulation (two-way mixed ANOVA; interaction band vs frequency, $F_{8,56} = 2.59$, p < 0.05; main effects of frequency band, $F_{2,56} = 54.02$, p < 0.0001 and stimulation frequency, $F_{4,56} = 3.3$, p < 0.05). Stimulation frequency of 0 indicates no stimulation. Error bars show SEM. *X* symbol indicates a significant interaction between factors.

value with the number of licks normally detected during the same period in non-stimulation trials. Stimulation would result in some additional licking even after offset of stimulation, but this was not related to the stimulation frequency (*Figure 4D*; two-way mixed ANOVA; main effect of stimulation, $F_{1,23} = 18.1$, p<0.001; main effect of frequency, $F_{3,23} = 3.58$, p=0.30; interaction, $F_{3,23} = 1.91$, p=0.15). These results show that greater stimulation of the direct pathway reduces licking initiation latency, and that licking continues for a short time beyond the immediate period of laser stimulation.

In contrast to D1 activation, indirect pathway activation in the VLS in well-trained mice resulted in a pronounced suppression of licking. When stimulation occurred 5 s prior to reward delivery (*Figure 5A*), we observed a reduction in licking rate (*Figure 5B and C*). We found a significant effect of experimental group on the change in licking rate (*Figure 5D*), two-way ANOVA, $F_{1,40} = 12.35$, p<0.01, but no effect of frequency ($F_{3,40} = 2.04$, p=0.12) nor an interaction between group and frequency ($F_{3,40} = 2.2$, p=0.1). Although we did regularly observe a suppression of licking activity 5 s prior to reward, the likelihood of licking at this period in the interval was low, reflecting a potential floor effect. When activating the indirect pathway 1 s prior to reward (*Figure 5E*), during peak anticipatory licking, we also observed a significant impact on licking by indirect pathway stimulation (*Figure 5F and G*, *Video 3*). In addition to a strong suppression of licking by indirect pathway stimulation (*Figure 5H*), two-way ANOVA; significant effect of experimental group $F_{1,42} = 35.48$, p<0.0001), we observed a significant effect of stimulation frequency on licking ($F_{3,42} = 8.88$, p<0.0001), and a significant interaction between group and frequency ($F_{3,42} = 6.31$, p<0.01). Post-hoc tests revealed significant suppression of licking by 25 Hz (p<0.001) and 50 Hz (p<0.001) stimulation. We were also able to suppress consummatory licking that occurred following reward (*Figures 5I, J and K*). There was a significant effect of experimental group on the change in lick rate (*Figure 5L*, two-way ANOVA, $F_{1,42} = 6.48$, p<0.05), a significant effect of frequency ($F_{3,42} = 4.01$, p<0.05), and a significant interaction between group and frequency ($F_{3,42} = 3.38$, p<0.05). Post-hoc tests showed that the interaction was mediated by significant suppression of licking by 25 Hz (p<0.05) and 50 Hz (p<0.05) stimulation. The effect of indirect pathway stimulation was also dependent on the time of stimulation. A two-way ANOVA revealed significant effects of both time of stimulation in the task (Figure 5M, $F_{3,76} = 21.38$, p<0.0001) and laser frequency ($F_{2,76} = 3.19$, p<0.05), with no significant interaction ($F_{6,76} = 2.1$, p=0.06). Together, these results reveal a frequency-dependent effect of indirect pathway stimulation on licking behavior. Furthermore, indirect pathway activation suggests less of an effect on licking occurring during consumption than on licking during anticipation.

When we activated the indirect pathway at 5 s prior to reward delivery, when mice were least likely to lick, we observed a striking rebound of licking following the offset of stimulation

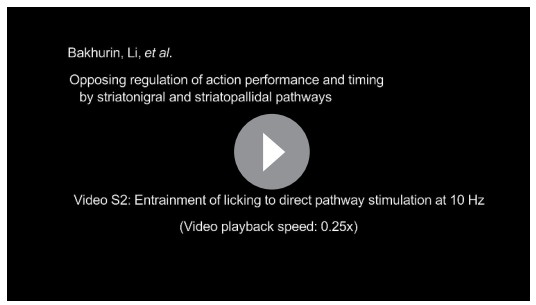

**Video 2.** Increase of licking frequency to 10 Hz with direct pathway stimulation. Video shows anticipatory licking behavior in a head-fixed D1-Cre mouse with ChR2 expressed in VLS. The video demonstrates that licking frequency can be boosted to close to 10 Hz with direct pathway stimulation, also at 10 Hz. Note the nearly one-to-one correspondence of laser pulses and licking. Stimulation begins one second prior to reward, lasts 1 s, and ends with reward. Stimulation is denoted with a blue square in the top right corner. Video playback is at quarter-speed.
https://elifesciences.org/articles/54831#video2

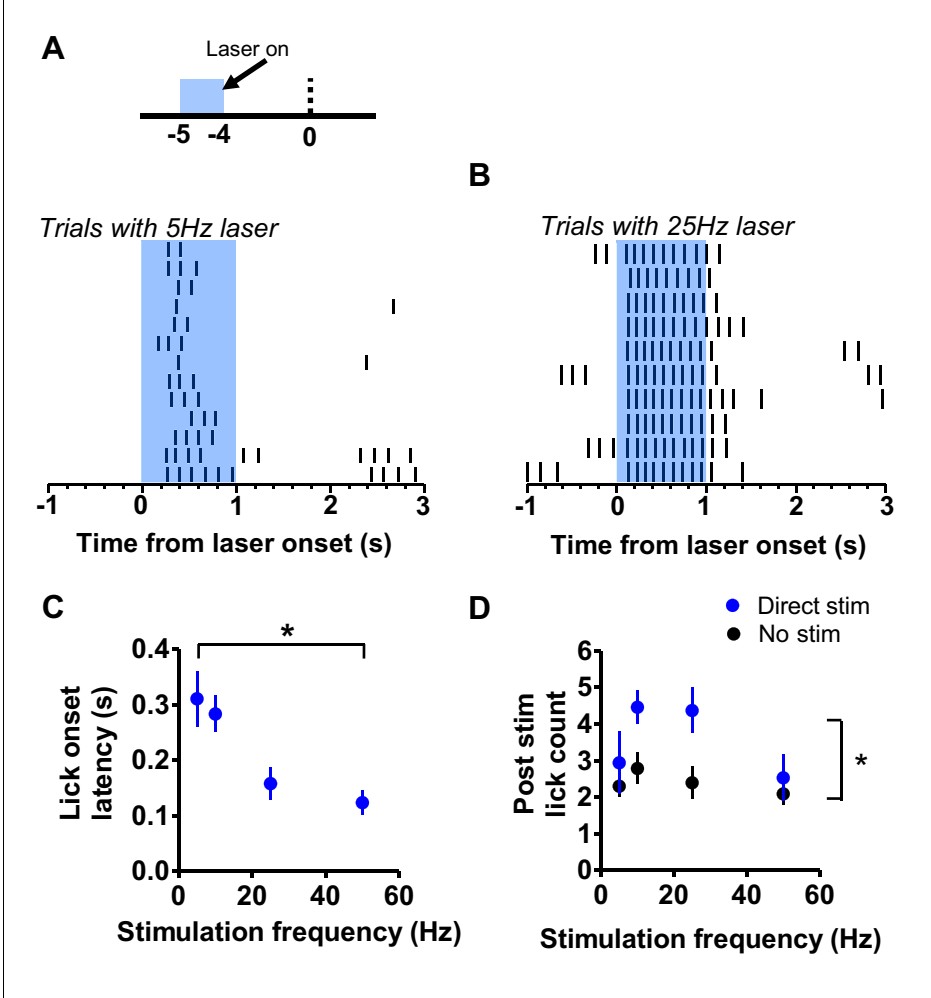

**Figure 4.** Direct pathway modulates onset latency and duration of evoked licking bout. (A) *Top*: The effect of stimulation frequency on evoked licking was measured for sessions with stimulation occurring at 5 s prior to reward. *Bottom*: Example raster plot showing longer latency to lick and shorter bout duration during 5 Hz direct pathway stimulation. B) Stimulation of the direct pathway at 25 Hz resulted in rapid licking onset and sustained licking following offset of laser stimulation. (C) Latency to lick bout onset as a function of the frequency of direct pathway stimulation (n = 8; one-way ANOVA; $F_{3,22}$ = 4.92, p<0.01). (D) Number of licks that were counted during the 1 s following direct pathway stimulation offset. Blue points reflect counts during trials with laser stimulation at 5 s before reward. Black points reflect the baseline count number observed during trials without stimulation for the same time period of the interval (two-way mixed ANOVA; main effect of stimulation: $F_{1,23}$ = 18.1, p<0.001; interaction stimulation vs. frequency, $F_{3,23}$ = 1.91, p=0.15). Error bars show SEM.

(*Figures 4B C*, *6A B*, *Video 4*). Lick rates during this rebound were higher during the second following laser offset as compared to the same time point in the absence of stimulation (*Figure 6C*, two-way mixed ANOVA, main effect of stimulation, $F_{1,24}$ = 11.08, p<0.01), but there was no main effect of frequency ($F_{3,24}$ = 0.17, p=0.9) and no interaction between stimulation frequency and the rebound licking rate ($F_{3,24}$ = 0.36, p=0.78). Indirect pathway stimulation also resulted in a reduced latency to initial lick onset following stimulation when compared to non-stimulation trials (*Figure 6D*, two-way mixed ANOVA, $F_{1,24}$ = 16.41, p<0.001; main effect of frequency, $F_{3,24}$ = 2.85, p=0.05). There was a significant interaction between trial type and the stimulation frequency, with higher frequencies resulting in lower onset latencies ($F_{3,24}$ = 5.73, p<0.01). Post-hoc tests showed that the interaction was mediated by shorter rebound latencies following 25 Hz (p<0.05) and 50 Hz (p<0.001) stimulation. Varying stimulation frequencies resulted in differing effects on the regularity of rebound latencies, indicated by measuring the variance of lick onset (*Figure 6E*, two-way mixed ANOVA, main

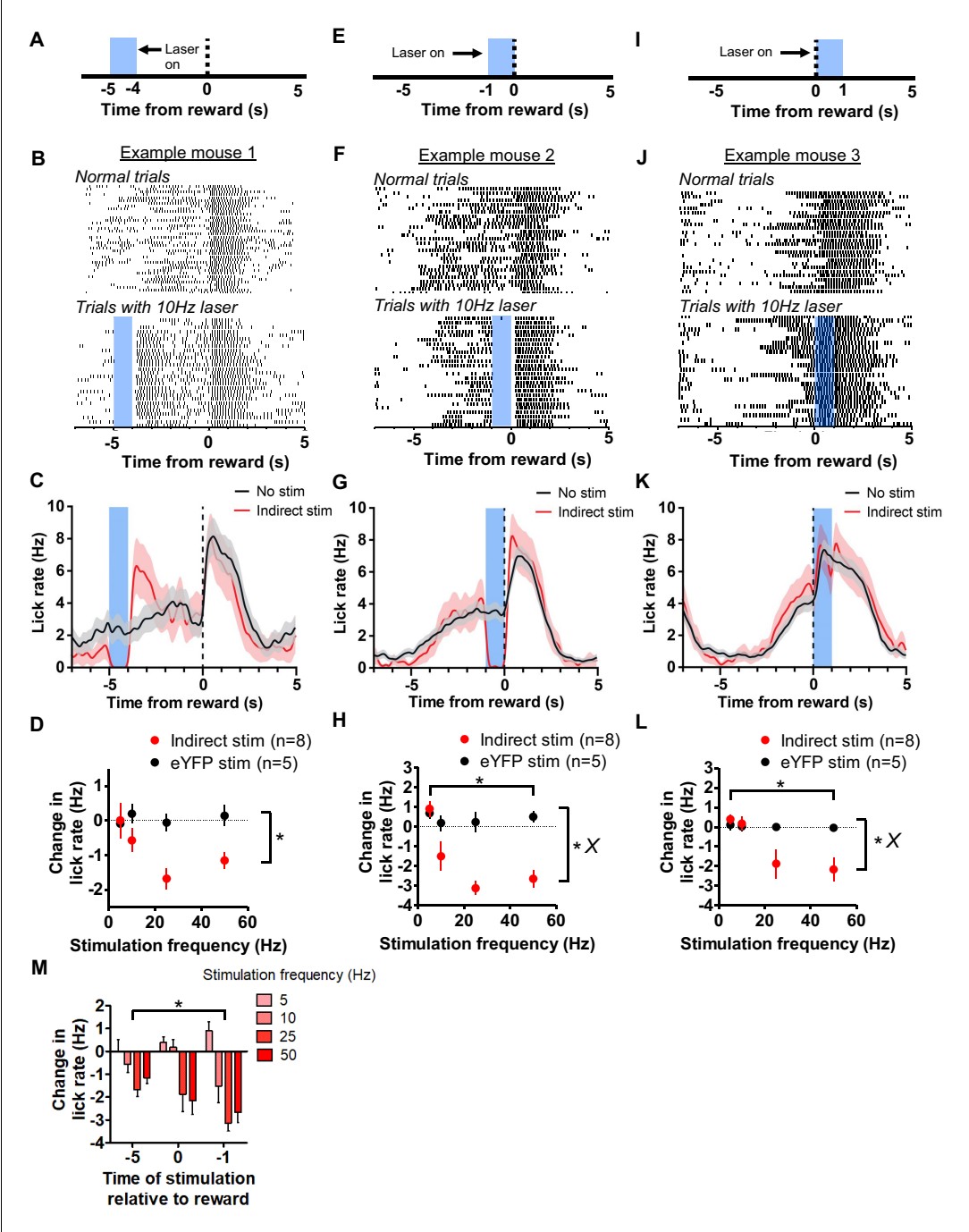

**Figure 5.** Stimulation of the indirect pathway in the orofacial striatum suppresses licking. (**A**) Depiction of stimulation strategy 5 s prior to reward in A2A-cre mice. (**B**) *Top*: Peri-event lick raster diagrams of representative trials demonstrating normal anticipatory licking patterns in trials without laser. *Bottom*:10 Hz stimulation at 5 s prior to reward results in a pause in licking. Both rasters reflect licking from the same mouse in the same session. (**C**) Mean lick rate calculated from the example session shown in *B* for trials with and without laser stimulation. (**D**) Mean reduction of licking rate during stimulation of the indirect pathway at 5 s (n = 8 A2A-ChR2 mice, n = 5 control mice; two-way ANOVA; main effect of experimental group, $F_{1,40}$ = 12.35, p<0.01). (**E**) Depiction of stimulation strategy 1 s prior to reward. (**F**) *Top*: Peri-event lick raster diagrams of representative non-laser trials. *Bottom*: Raster diagram of trials containing 10 Hz stimulation of the indirection pathway during anticipatory licking. Both rasters show licking from the same mouse in the same session. (**G**) Mean lick rate calculated from the example session shown in *F* for trials with and without laser stimulation. (**H**) Increasing stimulation frequency of the indirect pathway at 1 s prior to reward reduced the rate of anticipatory licking in A2A-Cre mice (two-way ANOVA; main effect of group, $F_{1,42}$ = 35.48, p<0.0001; main effect of frequency, $F_{3,42}$ = 8.88, p<0.0001; interaction group vs frequency, $F_{3,42}$ = 6.31, p<0.01). (**I**) Depiction of stimulation strategy coinciding with reward delivery. (**J**) *Top*: Representative lick raster diagrams of licking behavior during trials without laser stimulation. *Bottom*: Raster diagram of trials from the same mouse containing 10 Hz stimulation during consumption licking. Both rasters

*Figure 5 continued on next page*

*Figure 5 continued*

reflect behavior recorded in the same session. (**K**) Mean licking rate calculated from the example session shown in *J* for trials with and without stimulation. (**L**) Indirect pathway stimulation at the time of reward reduced licking rate (two-way ANOVA; main effect of group, $F_{1,42}$ = 6.48, p<0.05; main effect of frequency, $F_{3,42}$ = 4.01, p<0.05; interaction group vs frequency, $F_{3,42}$ = 3.38, p<0.05). (**M**) The time of stimulation during the interval influenced reductions in licking rate by stimulation (two-way ANOVA, main effect of time of stimulation, $F_{3,76}$ = 21.38, p<0.0001; main effect of Frequency, $F_{2,76}$ = 3.19, p<0.05). Error bars show SEM. *X* symbol reflects a significant interaction between factors.

effect of frequency, $F_{3,24}$ = 6.12, p<0.01; main effect of stimulation, $F_{1,24}$ = 3.3, p=0.08; interaction between stimulation frequency and trial type, $F_{3,24}$ = 8.63, p<0.001). Post-hoc tests showed greater variance of lick onset time following 5 Hz stimulation (p<0.05), and significantly lower variance following 50 Hz stimulation (p<0.01).

We observed that the effects of laser stimulation would be altered over the course of the behavioral session, as a function of motivational state. For example, the effects of direct pathway stimulation at 5 s prior to reward (*Figure 7A*) on changes in lick rate were often dependent on how many trials the mice received (*Figure 7B*). At higher frequencies, this gradual reduction of laser-generated licking rate was not as evident (*Figure 7C*). We divided the session into quartiles and quantified the increase of licking rate by stimulation as a function of trial quartile (*Figure 7D*). A two-way, mixed ANOVA revealed a significant interaction between frequency and quartile ($F_{9,72}$ = 2.62, p<0.05) in addition to significant effects of frequency ($F_{3,72}$ = 9.5, p<0.001) and trial quartile ($F_{3,72}$ = 3.64, p<0.05) on lick rate. Post-hoc tests showed significant reductions in lick rate between the 4th quartile and the 1st (p<0.001), 2nd (p<0.001), and the 3rd (p<0.01) quartiles under 10 Hz stimulation. We also found that the latency to stimulation-evoked lick onset increased over time. A two-way, mixed ANOVA revealed a significant effect of trial quartile on lick onset latency (Figure 7E, $F_{3,60}$ = 3.25, p<0.05) and a significant effect of frequency ($F_{3,60}$ = 5.67, p<0.01), but no interaction between trial quartile and stimulation frequency ($F_{9,60}$ = 0.38, p=0.9), suggesting a uniform increase in latency across all parameters. Like the effects of stimulation of the direct pathway on latency to lick, the latency to rebound licking following stimulation at 5 s was affected by motivation (*Figure 7F*). The latency of rebound licking increased as the mouse became sated in the course of a session, but this was most evident at higher stimulation frequencies (*Figure 7G*, two-way mixed ANOVA, interaction between trial quartile and stimulation frequency $F_{9,72}$ = 2.16, p<0.05; main effect of frequency, $F_{3,72}$ = 3.88, p<0.05; main effect of trial quartile $F_{3,72}$ = 4.04, p<0.05). Post-hoc tests showed that latency to rebound during the 4th quartile was significantly greater than during the 1st and 2nd quartiles under 25 Hz (p<0.05) and 50 Hz stimulation (p<0.01). These results suggest that engagement of lower-level licking generators by the BG can be modulated by motivational signals.

Our task allowed for the interrogation of the roles of the BG pathways in timing, as mice will readily anticipate the arrival of the reward (*Figure 8A*). We incorporated probe trials on which the reward was omitted to assess internal timing (*Figure 8B*). All mice generated anticipatory licking bouts during probe trials. Averaging across many trials results in a distribution of licking rates whose peak reflects the expected time of reward delivery, in this task at approximately 10 s. On a fraction of the probe trials we stimulated direct and indirect pathways to examine their contributions to timing.

We found that stimulation of the direct pathway during the probe trial could shift the peak rightward by an entire interval-duration. Thus, stimulation of the direct pathway concurrently with reward did not result in an obvious peak shift (*Figure 9A*, *left*). However, as we moved the stimulation time rightward by 3 or 5 s, the

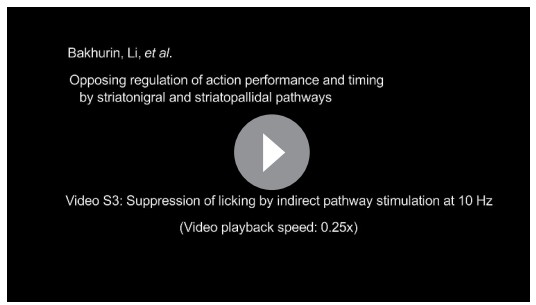

**Video 3.** Suppression of licking by indirect pathway stimulation at 10 Hz. Anticipatory licking behavior is shown in a head-fixed A2A-cre mouse with ChR2 expressed in VLS. Video shows rapid suppression of licking when stimulation is activated 1 s prior to reward. Stimulation is 10 Hz for 1 s, beginning 1 s prior to reward. Stimulation is denoted with a blue square in the top right corner. Video playback is at quarter-speed.

https://elifesciences.org/articles/54831#video3

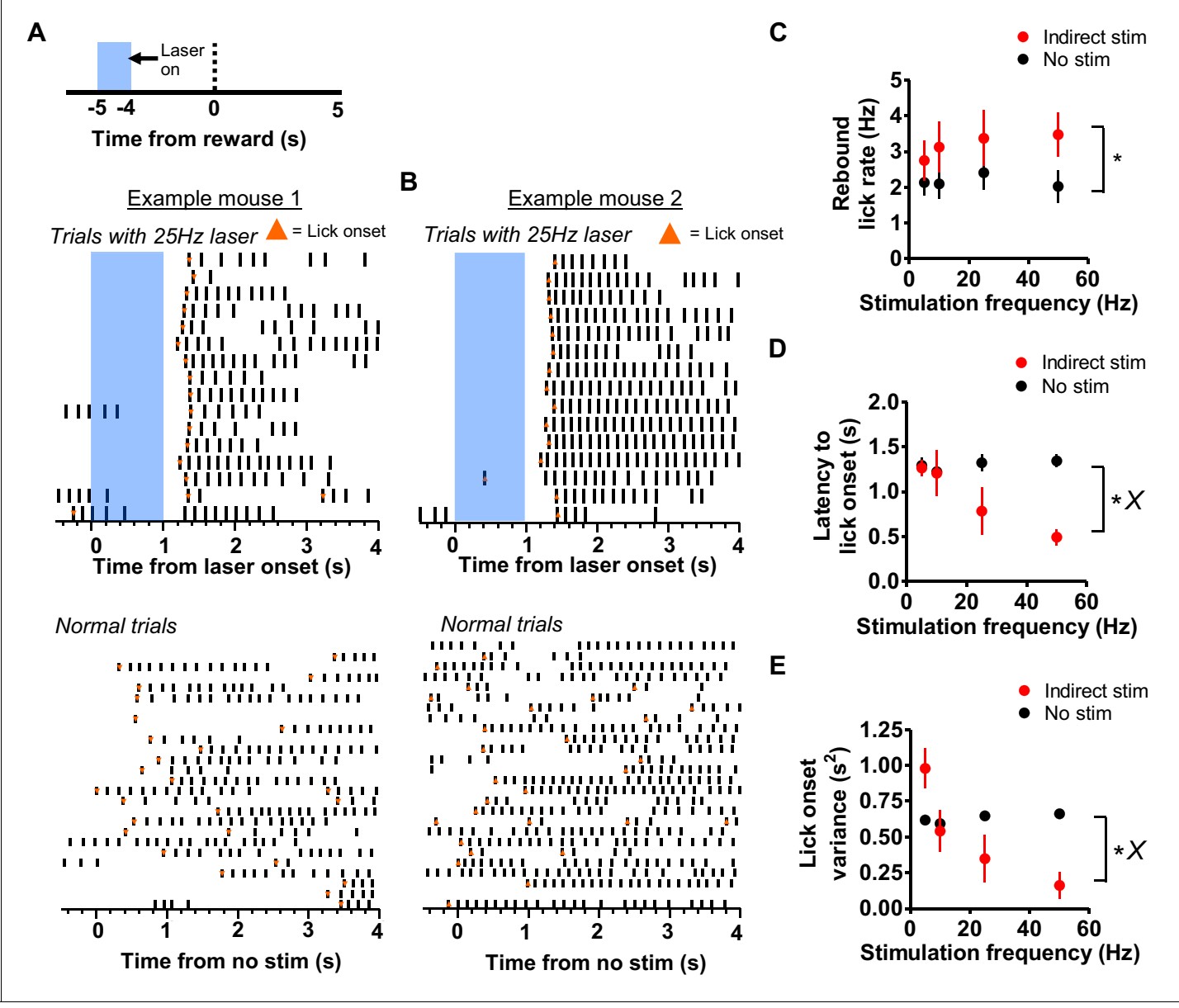

**Figure 6.** Rebound licking following stimulation of the indirect pathway. (**A**) *Top:* Schematic showing stimulation of A2A-Cre mice 5 s prior to reward. *Middle*: Example raster plot showing the resulting rebound licking following the termination of stimulation of the indirect pathway. Orange markers indicate the onset time of discrete lick bouts. *Bottom*: Raster plot of licking during no-stimulation trials from the same mouse, aligned to the same time point in the interval as above. Note the predominant alignment of lick onset times following laser stimulation. (**B**) *Top*: Example of another mouse showing rebound licking after indirect pathway stimulation. *Bottom*: Raster plot of licking during non-stimulation trials from the same mouse in the same session. C) Comparison of licking rate during stimulation and non-stimulation trials. Indirect pathway stimulation resulted in an increase in licking rate following laser termination when compared to the same time period in non-stimulation trials (n = 8 mice; two-way mixed ANOVA, effect of stimulation $F_{1,24} = 11.08$, p<0.01). (**D**) Reduction of lick onset latency following stimulation termination compared to mean latency for the same time period in non-stimulation trials (two-way mixed ANOVA, main effect of stimulation: $F_{1,24} = 16.41$, p<0.001; interaction stimulation vs trial type, $F_{3,24} = 5.73$, p<0.01). (**E**) Reduction of lick onset latency variance following stimulation offset compared to the variance of licking initiation for the same time period in non-stimulation trials (two-way mixed ANOVA, main effect of frequency, $F_{3,24} = 6.12$, p<0.01; interaction frequency vs. trial type, $F_{3,24} = 8.63$, p<0.001). Error bars show SEM. *X* symbol reflects a significant interaction between factors.

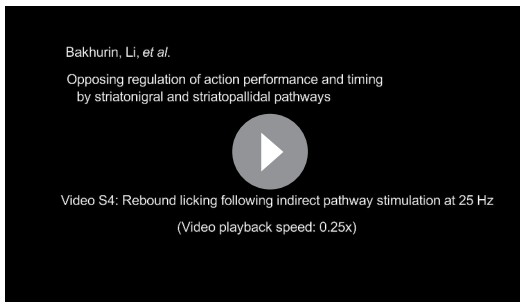

**Video 4.** Rebound licking following indirect pathway stimulation at 25 Hz. Video demonstrates the rebound licking effect of indirect pathway stimulation 5 s prior to reward, when mice are least likely to produce licking naturally. Stimulation is 25 Hz for 1 s and is indicated by the blue square in the top right corner. Video playback is at quarter-speed.

https://elifesciences.org/articles/54831#video4

peak would move rightward proportionally (*Figure 9A*, *middle* and *right*). Indeed, we found that the peak shift is a function of the start time of direct pathway stimulation (*Figure 9B*, one-way, RM ANOVA; $F_{3,9} = 21.68$, p<0.001). Control mice did not display any shifts in peak timing (data not shown; one-way RM ANOVA; $F_{2,8} = 0.7$, p=0.5). We also quantified other characteristics of the peak distributions that have been shown to be affected by pharmacological manipulations of the dopaminergic system. We found that direct pathway stimulation reduced the peak width, but we did not observe a significant effect of stimulation time on the width (*Figure 9C*, two-way mixed ANOVA; main effect of stimulation, $F_{1,12} = 35.49$, p<0.0001; no main effect of stimulation time, $F_{3,12} = 0.23$, p=0.8; no interaction between stimulation time and stimulation, $F_{3,12} = 1.43$, p=0.3). Direct pathway stimulation did not significantly impact the skew of the peak distributions (*Figure 9D*). These results reveal a critical role of direct pathway activation in resetting the internal timekeeping mechanism. This is the first report of a specific neural correlate of timer resetting. Interestingly, stimulation prior to reward did not impact the peak, suggesting that normally reward receipt may be involved in resetting the timing mechanism.

We also tested the effect of indirect pathway stimulation during probe trials. As observed previously, stimulation of the indirect pathway at 5 s after reward resulted in a rebound in licking following the termination of stimulation. This was also the case during peak probe trials. Mice showed different types of responses following rebound licking. First, rebound licking would result in a brief delay in licking which was followed by a bout of licking resembling a peak response ('recovery' pattern, *Figure 10A*). Moreover, rebound licking could also initiate the peak, so that the peak lasts longer than usual ('initiation' pattern, *Figure 10B*). Occasionally mice also generated a peak without showing a clear rebound following stimulation ('no-rebound' pattern, *Figure 10C*). These different patterns (recovery, initiation, and no-rebound) were equally common (*Figure 10D*, n = 5, one-way RM ANOVA; $F_{2,8} = 0.6$, p>0.05). Interestingly, peak analysis on these three licking patterns revealed distinct effects of indirect pathway stimulation on interval timing. For each pattern of licking, we compared the mean peak time with the peak time on trials in the same session that did not contain laser stimulation. During recovery-pattern trials, we found a large rightward shift of the peak that averaged approximately 2 s (*Figure 10E*, two-tailed t-test, $t_4 = 6.5$, p<0.01). In contrast, we found that when licking was initiated by the rebound licking bout, the peak was not affected (two-tailed t-test, $t_4 = 2$, p=0.1). On trials that lacked rebound licking, we found a rightward peak shift of approximately 750 ms (two-tailed t-test, $t_4 = 4.35$, p<0.05). Although initiation-type trials did not show a change in peak timing, we found that these peaks showed increased duration (*Figure 10F*, two-tailed t-test, $t_4 = 4.18$, p<0.05). We did not find a significant impact on the skew of the peaks in any trial type (*Figure 10G*).

Finally, we tested whether behavior following indirect pathway stimulation was altered as the session progressed. The probability of the recovery pattern decreased as the session progressed, whereas the likelihood of detecting no-rebound patterns increased (*Figure 10H*, two-way, mixed ANOVA; interaction of the effects of trial type and session half, $F_{1,12} = 7.38$, p<0.01). We did not detect a change in the probability of detecting the initiation-type pattern.

## Discussion

We found for the first time that optogenetic activation of direct-pathway neurons in the VLS can initiate licking (*Figure 2*), whereas activation of the indirect pathway in this region can terminate licking (*Figure 4*). These results are in accord with previous work implicating the VLS in orofacial behaviors

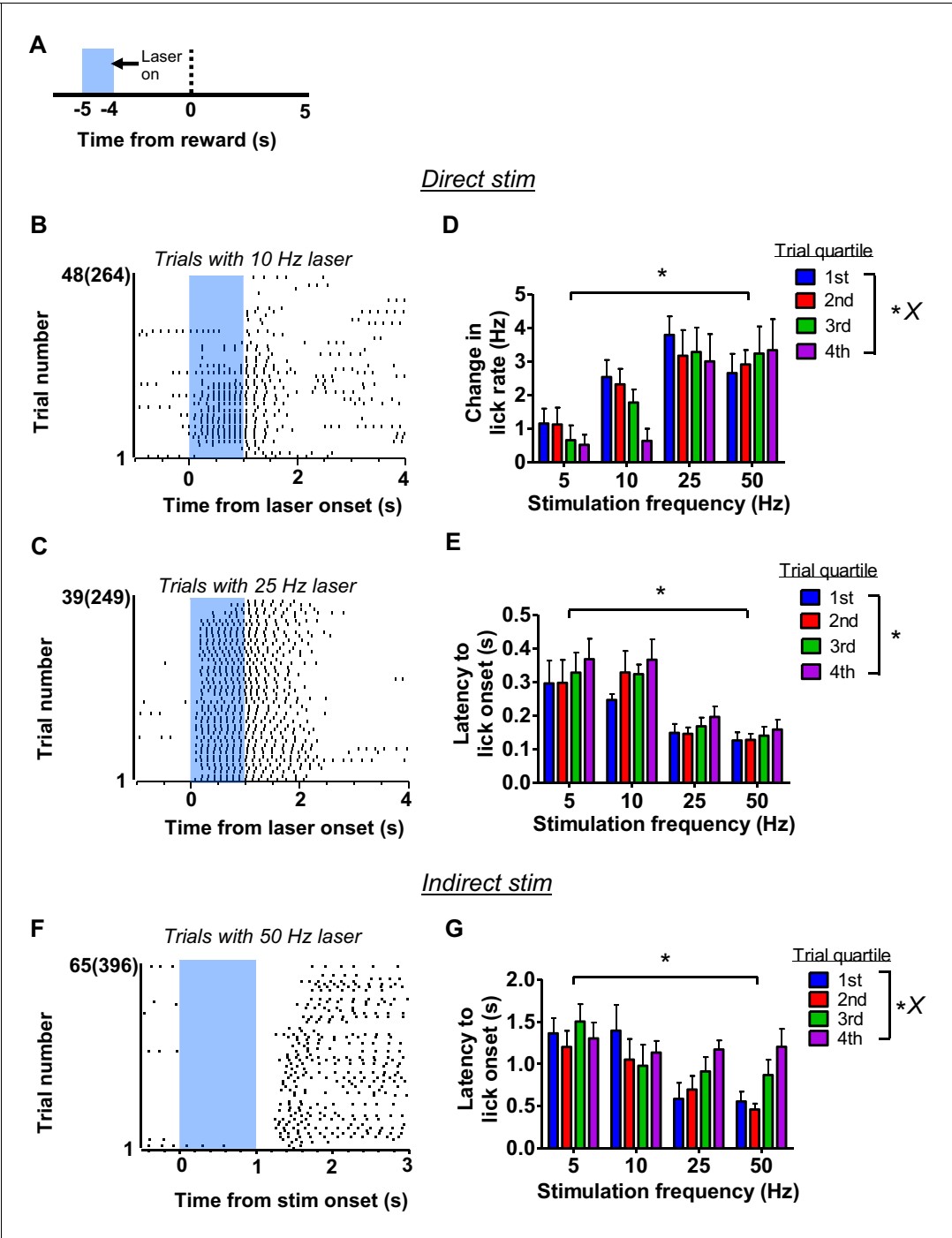

**Figure 7.** The effect of motivation on licking affected by direct and indirect pathway stimulation. (**A**) The effect of motivation on licking related to direct and indirect pathway manipulation was measured for sessions with stimulation occurring at 5 s prior to reward. (**B**) Example raster plot showing progressive reduction of licking evoked by 10 Hz direct pathway stimulation at 5 s before reward. Trial numbers reflect the number of laser presentations delivered with the number of rewards the animal received in parentheses. (**C**) Example raster plot showing a reduced influence of motivation on evoked licking with higher frequency stimulation of the direct pathway. Data are from the same mouse but a different session as shown in *B*. Trial numbers reflect the number of laser presentations delivered with the number of rewards the animal received in parentheses. (**D**) Grouped bar plot showing the reduced impact of laser stimulation at 5 s before reward on evoked licking as a function of trial quartile (n = 8; two-way mixed ANOVA; effect of quartile, $F_{3,72}$ = 3.64, p<0.05; effect of frequency, $F_{3,72}$ = 9.5, p<0.001; interaction quartile vs frequency, $F_{9,72}$ = 2.62, p<0.05). (**E**) Grouped bar plot showing an increase of lick onset latency as a result of direct pathway stimulation as a function of trial quartile (two-way mixed ANOVA, effect of quartile, $F_{3,60}$ = 3.25, p<0.05; effect of frequency, $F_{3,60}$ = 5.67, p<0.01; interaction quartile vs frequency, $F_{9,60}$ = 0.38, p=0.93). (**F**) Example raster plot showing the increase in rebound latency following 50 Hz stimulation of the indirect pathway over the course of the experimental

*Figure 7 continued on next page*

*Figure 7 continued*

session. Trial numbers reflect the number of laser presentations delivered with the number of rewards the animal received in parentheses. (**G**) Grouped bar plot showing the increase of lick onset latency following laser stimulation offset as a function of trial quartile (n = 8, two-way mixed ANOVA, effect of quartile, $F_{3,72}$ = 4.04, p<0.05; effect of frequency, $F_{3,72}$ = 3.88, p<0.05; interaction quartile vs frequency, $F_{9,72}$ = 2.16, p<0.05). Error bars show SEM. *X* symbol reflects a significant interaction between factors.

(*Mittler et al., 1994*; *Pisa, 1988a*; *Salamone et al., 1993*). The observed effects are distinct from those following activation of the dorsolateral striatum, which produces locomotion and turning behavior (*Bartholomew et al., 2016*; *Kravitz et al., 2010*; *Tecuapetla et al., 2014*). The acute effects on licking is not surprising given the topographical organization of the sensorimotor striatum and our target in the ventrolateral orofacial region.

Although BG projections to lower brainstem pattern generators are not required for reflexive orofacial and tongue movements (*Grill and Norgren, 1978*), intact BG are required for goal-directed food seeking behavior (*Bignall and Schramm, 1974*). It has been shown that a subset of GABAergic projections from SNr to the superior colliculus is critical for the initiation and termination of self-paced licking behavior (*Redgrave et al., 1992*; *Rossi et al., 2016*; *Toda et al., 2017*). A pause in licking can be caused by increased activity in nigrotectal output (*Rossi et al., 2016*; *Toda et al., 2017*). The GABAergic output from the SNr can also directly innervate the reticular formation (*Deniau and Chevalier, 1992*), which contains regions that are known to be involved in the generation of licking behavior (*Travers et al., 1997*). Direct pathway activation may reduce SNr output and indirect pathway activation may increase SNr output. Consequently, brainstem or tectal targets that receive GABAergic SNr projections may be disinhibited by direct pathway activation and suppressed by indirect pathway activation. We previously showed that opponent classes of SNr projection neurons command downstream position control systems in the midbrain and brainstem to move the

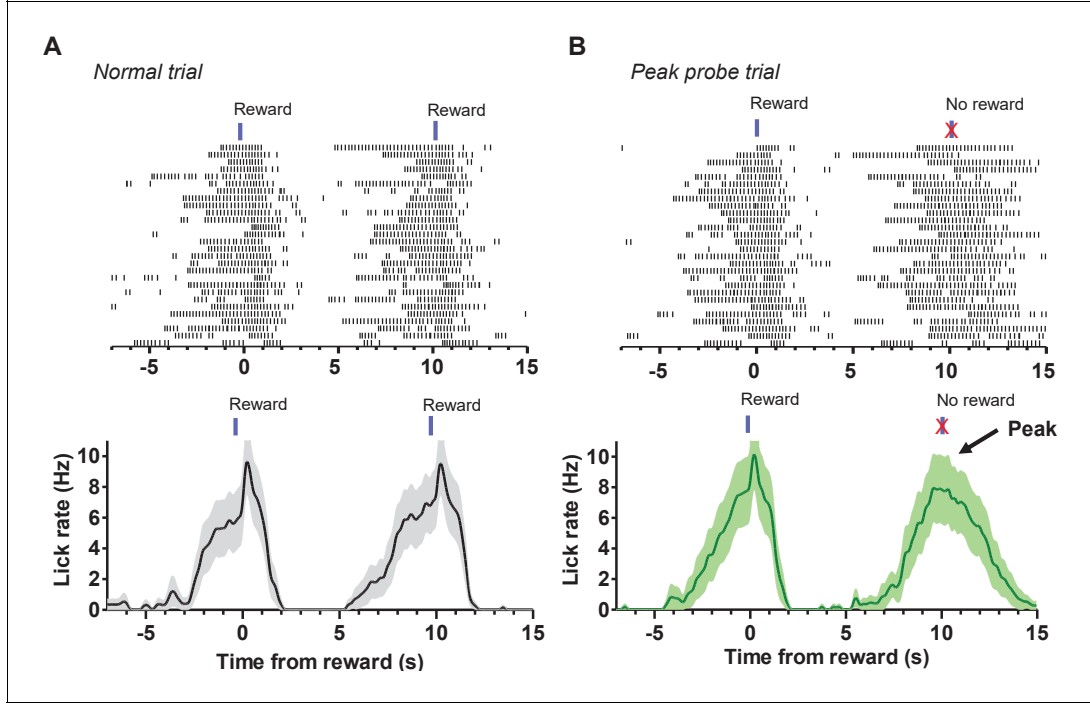

**Figure 8.** Peak probe trials reveal the operation of an internal timing mechanism. (**A**) *Top*: Example raster plot of licking during consecutive 10 s fixed-time trials. Licking is aligned to the first of two consecutive reward delivery times. *Bottom*: Mean licking rate for the session shown in the above raster. (**B**) *Top*: Example raster plot showing licking during peak probe trials. During peak trials, reward is delivered then withheld 10 s later, resulting in a discrete bout of licking in the absence of any stimuli. *Bottom*: The average licking rate for the session shown in the above raster. Averaging across probe trials results in a characteristic peak in licking. Data shown for both trial types were recorded from the same mouse and session. Error bars show SEM.

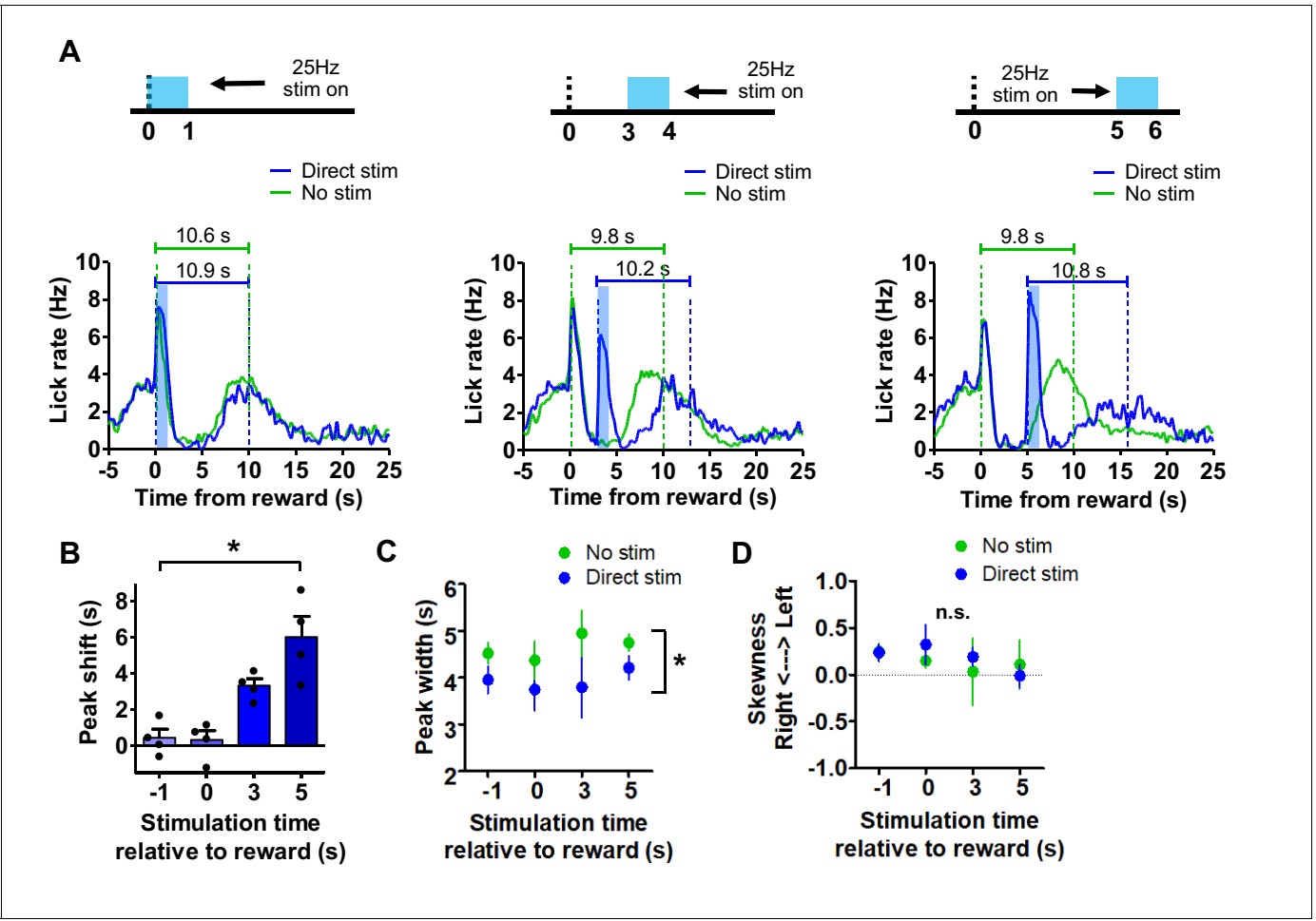

**Figure 9.** Direct pathway stimulation resets the internal clock. (A) *Left*: Mean licking rate across subjects during peak probe trials with and without laser stimulation concurrent with reward delivery. Scale bars reflect the population mean peak times for probe trails with (blue) and without (green) laser stimulation. *Middle*: Mean licking rate across subjects during probe trials delivered in the presence and absence of laser stimulation at 3 s post reward. Scale bars reflect the population mean peak times for probe trails with (blue) and without (green) laser stimulation. *Right*: Mean licking rate across subjects during normal probe trials and peak probe trials containing laser stimulation at 5 s following reward. Scale bars reflect the population mean peak times for probe trails with (blue) and without (green) laser stimulation. (B) Magnitudes of peak shifts in seconds as a function of time of direct pathway stimulation. The y-axis reflects the difference between the mean peak time occurring during laser trials subtracted by the mean peak time during trials without stimulation (n = 4 female mice; One-way RM ANOVA; $F_{3,9}$ = 21.68, p<0.001). Peak analyses were performed on the entire behavioral session to maximize statistical power. Points reflect individual data points. (C) Quantification of the peak duration during normal peak probe trials and those with laser stimulation. Stimulation resulted in a reduction of the peak width (n = 4; two-way mixed ANOVA; effect of stimulation, $F_{1,12}$ = 35.49, p<0.0001). D) There were no changes in the skewness of the peak distributions. Error bars show SEM.

body in different directions (*Barter et al., 2015b*; *Fan et al., 2012*; *Yin, 2014b*). The present results suggest a critical role of the direct and indirect pathways in modulating the lower level controllers for orofacial behaviors.

## Top-down regulation of lick pattern generator

A variety of evidence suggests that the output nuclei of the BG may function as an integrator (*Barter et al., 2015b*; *Yin, 2014a*; *Yin, 2017*). The rate of change in the integrator output is proportional to the input magnitude. With continuous control of position, a larger striatal signal (spikes per time window) is converted to faster rate of change in position control command (higher movement velocity) as well as greater movement amplitude (*Barter et al., 2015a*; *Bartholomew et al., 2016*). For licking behavior, we assume that the basal ganglia output also produces a top-down command from an integrator that influences the operation of the brainstem pattern generator. However, when the behavior in question is generated by pattern generators with relatively fixed innate rhythms, the

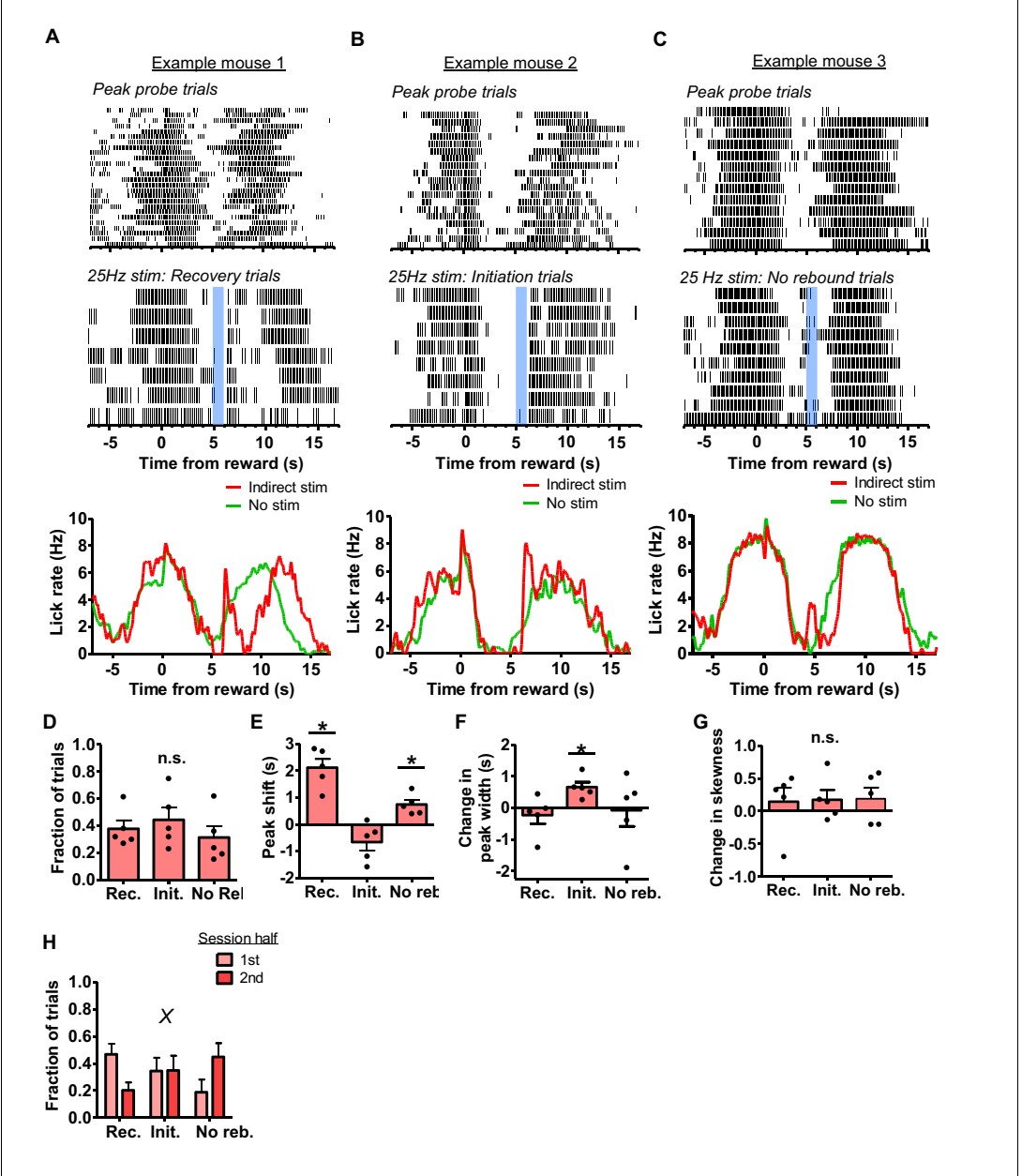

**Figure 10.** Indirect pathway stimulation pauses the internal clock. (A) *Top*: Peri-event lick raster of representative peak probe trials without laser delivery. *Middle*: 25 Hz stimulation at 5 s following reward during trials in which rebound licking causes a delay and a recovery in peak licking. Both rasters reflect licking from the same mouse in the same session. *Bottom*: Mean lick rate calculated from the example session shown above for peak probe trials without laser stimulation and stimulation trials showing a 'recovery' pattern. (B) *Top*: Peri-event lick raster diagrams of representative peak probe trials without laser delivery. *Middle*: 25 Hz stimulation at 5 s following reward during trials in which rebound licking initiates peak licking. Both rasters show licking from the same mouse in the same session. *Bottom*: Mean lick rate calculated from the example session shown above for peak probe trials without laser stimulation and stimulation trials showing an 'initiation' pattern. (C) *Top*: Peri-event lick raster of a third subject's representative peak probe trials produced in the absence of laser delivery. *Middle*: 25 Hz stimulation at 5 s following reward during trials in which no rebound licking was detected. *Bottom*: Mean lick rate calculated from the example session shown above for peak probe trials without laser stimulation and stimulation trials showing a 'no rebound' pattern. (D) The three types of patterns occurred with equal probability throughout the behavioral session (n = 5 (3 female and two male); one-way RM ANOVA, $F_{2,8}$ = 0.64, p=0.5). (E) Peak shifts were measured by subtracting the mean peak time of each pattern by the mean peak time without stimulation. Recovery and no rebound trials showed significant positive peak shifts (two-tailed t-tests, p<0.05), whereas initiation trials trended toward negative peak shifts (two-tailed t-test, $t_4$ = 2, p=0.1). (F) Initiation trials showed significant increases in the duration of the peak (two-tailed t-test, $t_4$ = 4.18, p<0.05). (G) There were no changes in skewness of licking distributions for any trial type. (H) Fraction of peak trials showing each pattern during the first and second halves of the behavioral session. On the whole, the recovery pattern was gradually

*Figure 10 continued on next page*

*Figure 10 continued*

replaced with no rebound pattern (two-way, mixed ANOVA, interaction trial type x session half, $F_{1,12}$ = 7.38, p<0.01). Error bars show SEM. *X* symbol reflects a significant interaction between factors. Points in bar graphs reflect individual data points.

mechanism is different. In this case, the BG output does not modulate the lick frequency in a continuous manner (*Rossi et al., 2016*), as it is not directly responsible for the generation of each lick. With continuous modulation, we would expect lick frequency to linearly scale with direct pathway stimulation frequency (*Figure 11*), but this was not observed. Once activated, the pattern generator appears to operate in several discrete settings, analogous to a fan with settings like off, low, and high. Normally, the lick pattern generator oscillates at 5–6 Hz for anticipatory licking and 6–8 Hz for consummatory licking. When the direct pathway was stimulated, licking could reach a new regime (~10 Hz) which appears to be the limit of the lick pattern generator and possibly also the biomechanical constraints of the orofacial musculature. When stimulation frequency exceeded 10 Hz, the licking frequency cannot increase further.

BG output may regulate the activity of the lower level lick generators. The integrator is filled by direct pathway activation. Because direct pathway projections are inhibitory, the integrator is filled by inhibiting GABAergic nigral neurons, which results in a disinhibition of downstream structures. Greater activation of the direct pathway produces longer suprathreshold activation of the pattern generator (*Figure 11B*). This could be responsible for the greater licking frequencies that we observed (*Figure 11C*). The increased power of licking in a given frequency band, indicating longer 'on time' or higher duty cycle of the oscillator, suggests the presence of the integrator (*Figures 3* and *11*). We also observed decreasing latencies to lick onset during higher stimulation frequencies, and that licking could be sustained for some time after the end of stimulation (*Figure 4*). Artificially driving the direct pathway efficiently fills the integrator, which results in saturation and maximum licking rates even at relatively low stimulation frequencies. However, mice rarely enter this regime on their own, perhaps due to the concurrent activity of the indirect pathway.

The indirect pathway, on the other hand, may play an important role by discharging the integrator, preventing the instantaneous licking frequency from reaching its maximum level in natural licking. Selective activation of the indirect pathway is equivalent to rapidly reversing the BG output signal, bringing or keeping the pattern generators below their activation thresholds and pausing behavior. It is therefore only under conditions of artificially activating the direct and indirect pathways separately that behavior can be described as operating in an all-or-none, go/no-go fashion. Prior observations of coincident activity of the two pathways may reflect coordinated regulation of behavior to keep output within a certain range (*DeLong, 1990*; *Tecuapetla et al., 2014*). The balanced relationship between these two pathways may be responsible for the continuous adjustments to specific kinematic details of ongoing movement.

We propose that the BG provide top-down modulation of central pattern generators in the form of duty cycle regulation. Coordinated activity of the direct and indirect pathways are combined to produce a BG output that modulates how long a relatively stereotyped pattern generator is activated. This form of top-down regulation is distinct from the more continuous regulation of position controllers (*Barter et al., 2015b*; *Yin, 2014b*).

We observed a reliable rebound of licking following indirect pathway stimulation in the middle of the interval (*Figure 6*). This was only possible to detect because mice rarely lick at that time point, and the effect was not masked by the delivery of reward or the presence of consummatory licking. In addition, higher frequency stimulation resulted in shorter latency to rebound licking as well as more uniform timing in the start of rebound licking. Indirect pathway stimulation may result in more active nigral output, which results in a large suppression of downstream structures. The release of this inhibition then gives rise to a burst of rebound activity responsible for lick generation, resulting in fast reactivation of licking CPGs above the threshold, and a more consistent latency of the rebound licking.

The effects of direct and indirect pathway stimulation were sensitive to the motivational state, as the latencies to licking, overall licking rates, and the latency to rebound all changed over the course of the session (*Figure 7*). This could not be explained as simply a reduction in the efficacy of laser stimulation over the course of a single session because stimulation of the direct pathway at higher frequencies could override the effects of reducing motivation. These results can be explained by the

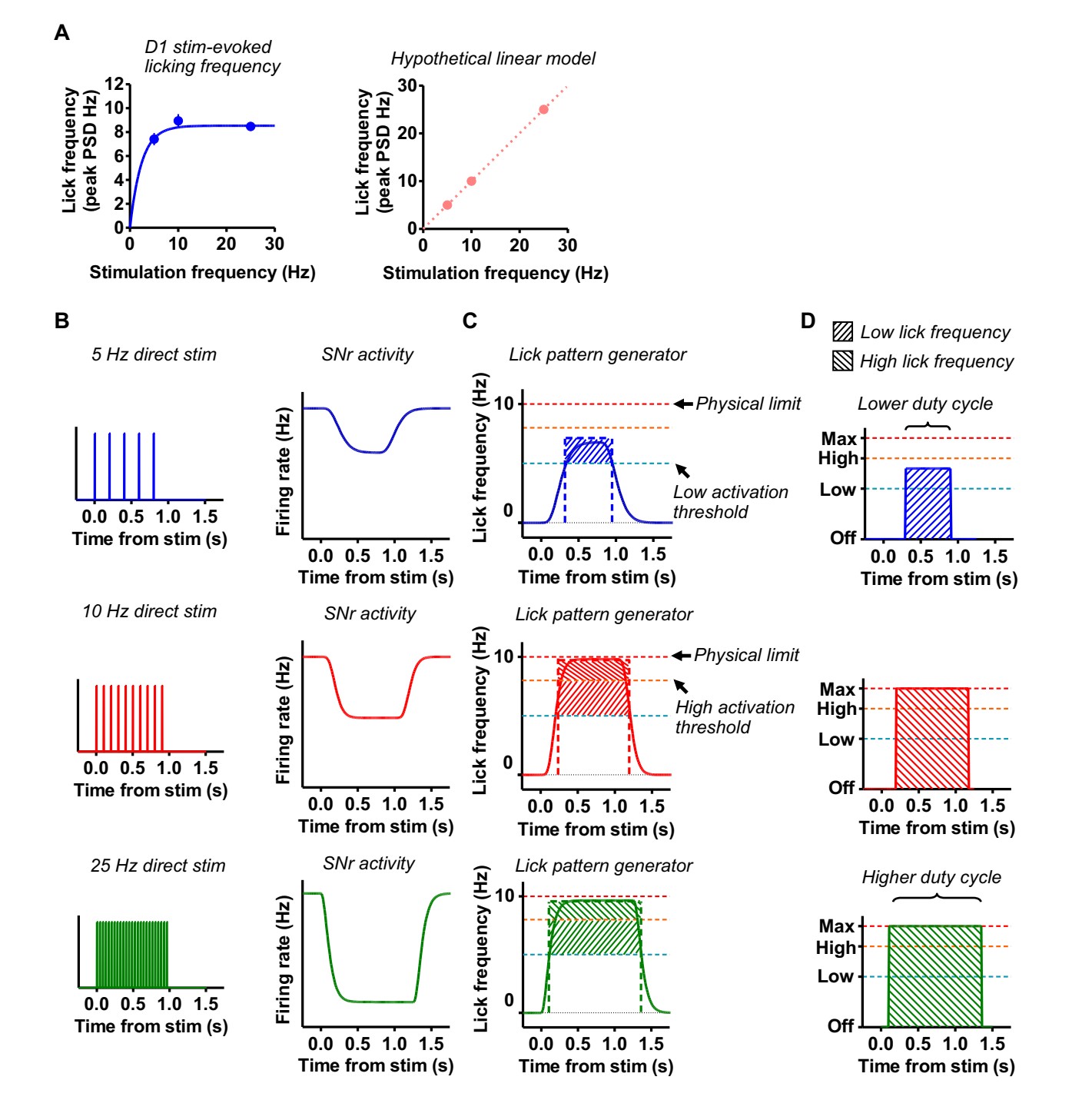

**Figure 11.** Direct pathway modulates licking CPG activity via integration. (**A**) *Left*: Laser-evoked licking frequency corresponding to the peak of the power spectral density distribution as a function of direct pathway laser stimulation frequency. Data for stimulation frequencies 5–25 Hz are shown. *Right*: Hypothetical expected linear laser-evoked licking frequencies if direct pathway stimulation directly drove each lick. Note that in this scenario, 5 Hz stimulation would result in licking at 5 Hz and 25 Hz stimulation would result in licking at 25 Hz. (**B**) Proposed mechanism translating direct pathway stimulation to SNr output activity via integration. Higher frequency direct pathway stimulation results in faster filling of the integrator that leads to a faster rate of change as well as longer-lasting output. (**C**) Increasing SNr output brings lower-level licking centers above several different activity thresholds, corresponding to varying discrete licking frequencies. 5 Hz stimulation does not result in 5 Hz licking. CPG output is capped at 10 Hz. (**D**) Greater filling of the integrator results in a more sustained licking output at a given frequency level. This is reflected in the increasing power in a given frequency band with increasing frequency of stimulation, suggesting a role for the modulation of licking duty cycle by the BG.

possibility that, over time, the activation threshold for licking increases, blunting the influence of descending signals.

The effects of stimulation on licking varied in a manner that was dependent on the time in the task when the stimulation occurred. Direct pathway stimulation had the greatest impact on licking when mice were not licking, in the middle of the interval. On the other hand, stimulation during ongoing licking had a smaller effect. This could be explained by a ceiling effect, as mice could only boost their licking frequency so much once they were already performing the behavior. The opposite trend was observed with indirect pathway stimulation: stimulation at 5 s prior to reward showed a lower change in rate than stimulation during anticipatory licking.

## Striatal contribution to timing

Our results also shed light on the distinct contributions of direct and indirect pathways to timing. The use of the fixed-interval schedule allowed us to incorporate peak probe trials that measure interval timing in mice.

Some have suggested that the brain's ability to time intervals is an emergent property of sequential neural population dynamics (*Bakhurin et al., 2017*; *Crowe et al., 2014*; *Mauk and Buonomano, 2004*; *Mello et al., 2015*). However, these models do not explain why specific neural circuits are critical for timing; nor do they incorporate detailed anatomical organization such as the direct and indirect pathways, which are shown to have distinct roles in timing. Our observations support classic pacemaker-accumulator models (*Gibbon et al., 1984*), which contain specific mechanisms that can be paused or reset (*Buhusi and Meck, 2002*). In fact, the proposed BG integrator mechanism is quite similar to the pacemaker-accumulator model, which also makes use of an integrator.

We discovered that stimulation of the direct pathway can reset the interval timing mechanism and activation of the indirect pathway could pause its operation (*Figures 9*, *10*). During peak probe trials, mice show peak responding at the trained interval of 10 s. When stimulating the direct pathway, we found that the peak would appear approximately 10 s following the onset of stimulation. The interval was maintained, but the activation of the direct pathway appeared to reinitiate the accumulation of the 10 s interval, just like resetting some 'internal clock'.

On the other hand, indirect pathway stimulation did not result in a resetting effect, suggesting that resetting is specific to direct pathway activation and not a consequence of any distracting neural perturbation. This also makes it unlikely that the rebound effect is mediated by excessive direct pathway activity. We found an interaction between the timing system and the presence of rebound licking immediately following indirect pathway stimulation. In general, the chief effect of indirect pathway activation appears to be a pause in the timer, similar to what has been observed previously with nigrotectal stimulation (*Toda et al., 2017*). Driving the indirect pathway paused the 'clock,' and the presence of rebound licking could extend the duration of the pause. This impact on timing was only observed occasionally, as sometimes rebound licking simply initiated peak responding. These trials were also noteworthy in that they resulted in a peak of longer duration, but not one with a significant leftward shift in the peak time. Thus only the precision of timing was affected. The distinct types of trials observed could be explained by variable rates of filling the accumulator: faster accumulation on a given trial would allow the rebound licking to initiate the peak, whereas with slower accumulation the indirect pathway activity could interrupt or pause this process. In partial support of this idea are our observations of the motivational effects on behavioral responses to indirect pathway stimulation. As rebound licking latency increases over time, the proportion of peak-probe trials that contained rebound licking also decreased.

## Conclusions

In summary, we show that direct pathway stimulation in the VLS can initiate licking, whereas indirect pathway stimulation can suppress ongoing licking, and that these pathways can work in concert to continuously regulate ongoing licking. In addition, the distinct effects of direct and indirect stimulation on timing behavior suggest that each pathway differentially interacts with the internal time-keeping system: direct pathway stimulation can reset the timer, whereas indirect pathway activation can only pause or interfere with it. The presence of distinct mechanisms for resetting and pausing an internal clock could be related to opponent roles of direct and indirect pathways in the integration process, akin to the accumulator in timing models. Together these results suggest for the first time a

uniform underlying mechanism that can explain the role of the BG circuits in action generation as well as interval timing.

Some have argued that the accumulator is distributed in the cortex (*Wang et al., 2018*). Our results, however, suggest that it resides in the BG output circuit, though the underlying mechanisms for integration remain unclear. Future work will be necessary to elucidate the neural implementation of this integrator and how the BG output influence and modulate targets in the midbrain and thalamus to regulate performance and timing of actions.

Finally, a few limitations of the current study should be noted. We cannot rule out that the same striatal units may be involved in multiple different behaviors depending on context. As the head-fixed preparation makes it impossible for the other behaviors to be expressed, the trained behavior, namely licking, may become the prioritized action under this condition. It would be important in future studies to examine the role of the VLS in unrestrained animals, and the relationship between orofacial behavior and other components of natural appetitive behaviors.

# Materials and methods

## Subjects

8 adult D1-Cre (4 male and 4 female) and 13 A2A-Cre (5 male and 3 female mice received ChR2; 2 male and 3 female mice received eYFP injections) mice of both sexes were used (Jackson Labs, Bar Harbor, ME). D1-Cre mice have Cre-recombinase targeted to the *Drd1a* locus, and A2A-Cre mice have Cre-recombinase targeted to the *Adora2a*.This enabled selective expression in neurons in the direct and indirect pathways, respectively. All experiments were conducted during the light phase of the 12:12 light cycle. Mice were housed in groups of two or three. Mice had unrestricted access to food in their home cages, but were water deprived and received a 10% sucrose solution during behavioral sessions. Body weight was maintained at 85% body weight and was monitored on a daily basis, with additional water given after experimental training as needed. All experimental procedures were approved by the Duke University Institutional Animal Care and Use Committee.

## Viral vectors

AAV5-EF1$\alpha$-DIO-hChR2(H134R)-eYFP was injected into 8 D1-Cre and 8 A2A-Cre mice. AAV5-EF1$\alpha$-DIO-eYFP was injected into 5 A2A-Cre mice for control experiments. Vectors were obtained from the University of North Carolina Vector Core.

## Surgery

Mice were placed in a stereotaxic frame and anesthetized with ~1–1.5% isoflurane. Small holes were drilled bilaterally at AP: +0.2 mm and ML: + / - 2.5 mm relative to bregma. Stereotaxically guided injections of (200 nL per hemisphere) were made at AP: +0.2 and ML: 2.5 and DV: −4.6 and −4.4 relative to bregma using a Nanoject III microinjector (Drummond Scientific, Broomall, PA). 100 μm-core optic fibers (Precision Fiber Products, Chula Vista, CA) were implanted with the same AP and ML coordinates and at DV: −4.2 in order to target cell bodies of the VLS. Fibers were secured to the skull with screws and dental cement along with a head post for head fixation.

## System for behavioral training and recording

Mice were allowed 7–14 days to recover from surgery and then were water deprived in their home cages. The interval timing task is the same as previously described (*Toda et al., 2017*). Briefly, behavioral experiments were conducted in a sound-proof chamber with the mice perched in a custom-made elevated tunnel platform. For head fixation, an implanted steel head post was clamped on both sides of the head. A metal drinking spout was positioned directly in front of the mouth so that the mouse only needs to protrude its tongue to access the spout. A 10% sucrose solution was gravity fed to the spout and its delivery was controlled by the opening of a solenoid valve. The spout was connected to a capacitance touch-sensor (MPR121, Adafruit, New York, NY) coupled to an Arduino Leonardo (www.arduino.cc) that reported the time of each contact with the spout by the tongue. The task was controlled using Matlab (version 2014b, Mathworks, Natick, MA) programs interfaced with a Blackrock Cerebus recording system (Blackrock Microsystems, Salt Lake City, UT) to generate digital signals to control reward delivery, and analog commands to drive laser

stimulation. The Cerebrus system was simultaneously used for recording these signals in addition to the digital timestamps received from the lickometer. All timestamps were saved for offline analysis.

## Fixed-interval timing task

Initial training began with water-deprived mice and involved habituation to head-fixation and training mice to collect experimenter-delivered water drops from the spout. Once reliable licking was established for drop delivery after a few days, training began on a fixed-interval version of the task. During the task, mice received a 5 µL drop of reward every 10 s. No external stimuli were presented. White noise was played inside of the chamber to mask the sound of the solenoid (*Rossi and Yin, 2015*). Water was presented at fixed intervals for 200 trials a day until mice showed anticipatory licking behavior.

## Peak probe trials

Once mice displayed consistent anticipatory licking in during fixed-interval sessions, sessions began to incorporate peak probe trials. During peak trials, no reward is presented in order to quantify the internal representation of time. Peak probe trials occurred with a 60% possibility after three consecutive rewards. No indication was given as to when a peak trial would occur. Peak trials lasted 30 s plus a random duration sampled from a gamma distribution (Matlab function *gamrnd* using shape parameter = 2.5 and scale parameter = 4). The consecutive reward counter was then reset with a reward delivery. Mice that struggled to learn this version of the peak procedure were run on a schedule that used a 30% probability of peak probe trial occurrence after three consecutive rewards.

## Optogenetic stimulation

Optical stimulation occurred both in rewarded and probe trials to prevent mice from predicting the probe trials with the sensation of the stimulation. Stimulation on a given trial was determined using random sampling from a uniform distribution. Following three fixed-interval trials without laser delivery, the program entered a decision point. There was a 30% chance of laser delivery during a normal trial, a 30% chance of laser delivery during a peak trial, and a 30% chance of a peak trial without laser. The remaining 10% of cases resulted in a new decision point. This resulted in a 10–15% of all trials being a peak trial without stimulation, a peak trial with stimulation, or a normal. Once head fixed in the testing chamber, mice were connected bilaterally to a 473 nm DPSS laser (BL473T3, Shanghai Laser, Shanghai, China) via fiber optic cables. A fiber splitter (TM105R5F1A, ThorLabs, Newton, NJ) divided the beam (50:50) for bilateral stimulation. Stimulation was pulsed (0.8–2 mW; 5–50 Hz, 10 ms square pulse width, 1 s duration). Stimulation onset was either at −1 s from reward, was concurrent with the time of reward (0 s condition) or occurred 5 s prior to the reward. During peak probe trials, stimulation was delivered at −1 s from reward, with the time of reward, or 5 s following reward delivery. Mice received multiple sessions with stimulation in order to account for the effects of stimulation frequency and timing relative to reward on licking, meaning that animals received at least 12 (3 times in the trial x four stimulation frequencies) sessions of stimulation, in addition to additional tests to investigate the effects on timing. Stimulation parameters (stimulation frequency and timing relative to reward) were consistent within a session, but the order of stimulation was semi-randomized between mice. Stimulation could be repeated over multiple consecutive sessions without detriment to timing (data not shown).

## Quantification and statistical Analysis

Behavioral data were analyzed with NeuroExplorer (Nex Technologies, Colorado Springs, CO) and Matlab. Licking timestamps were filtered to exclude events occurring less than 8 ms apart. Unless otherwise stated, all analyses were performed on the first half of trials to exclude motivational effects. Statistical tests were performed in GraphPad Prism (GraphPad Software, San Diego, CA). Two-tailed or one-tailed parametric tests were used. Post-hoc tests were done using Bonferroni correction. Power spectrum analysis was performed in NeuroExplorer.

## Peak analysis

Peak-detection and quantification was performed on a single-trial basis. For each peak probe trial, licking time stamps were aligned to the reward that initiated the trial. An analysis window extending

0 to 20 s following the reward was used to detect peaks. Licking within this window was binned using 100 ms time bins to produce a continuous licking rate estimate and smoothed using a gaussian filter with a width of 1 s. The Matlab function *findchangepts* was then applied to the smoothed lick rate signal to detect steep transitions in licking rate. These change points were classified as rising or falling transitions based on the slope of the lick rate signal around them. The first increasing change point in the analysis window and the next falling change point were determined to represent the start and end of the peak, respectively. The peak was fit with a gaussian function, the maximum of which was taken as the peak for that trial. Peak duration had to be at least 2 s in duration and licking rate had to exceed 3 Hz to be included. In cases of failures to identify a peak, the analysis window was shifted by +500 milliseconds and the procedure was applied again up to a maximum of 5 shifts, after which the trial would be discarded upon failure to detect a valid peak. To detect peaks occurring after direct pathway stimulation at 3 or 5 s, the 20 s analysis window was shifted right by 3 or 5 s, respectively, following the reward to exclude stimulation-evoked licking as a peak. Skewness was calculated from distributions of all lick-times occurring during peak responses across trials. Peak widths were calculated by subtracting the start-times of the peak from the end-times for each trial.

### Lick onset latency analysis

Lick onsets was defined as the first lick occurring after all inter-lick-intervals of 1 s or greater. For determining the latency to lick onset, the first lick onset occurring during the 1 s laser stimulation was counted for each stimulation trial. For determining rebound licking onset time, we measured the time until the first lick onset that occurred within a window of 0 to 3 s following the offset of laser termination. The same window was applied to non-laser trials for within-subject comparisons. Trials without licking were excluded from analysis.

### Post rebound peak trial classification

Peak trials that contained indirect pathway activation at 5 s post reward were classified by the temporal relationships of lick onsets relative to rebound licking. Trials that contained a lick onset in a 2 s period following stimulation onset were considered having a rebound. Otherwise, the trial was classified as lacking a rebound. For trials classified as a rebound-containing peak trial, a second lick onset timestamp had to occur within 8 s following initiation of rebound licking. In this case, the trial was classified as a rebound recovery trial. Otherwise, if no lick onset timestamp followed the rebound, the trial was classified as a rebound absorption trial. To detect peaks during rebound recovery trials, the analysis window was shifted by 6 s from reward delivery in order to exclude the rebound from being counted as a peak.

## Acknowledgements

This work was supported by NIH grants DA040701, NS094754 and MH112883 (HHY). We thank Dr. Fengxia Allen for technical assistance.

## Additional information

### Funding

| Funder | Grant reference number | Author |
| --- | --- | --- |
| National Institute of Mental Health | MH112883 | Henry Yin |
| National Institute on Drug Abuse | DA040701 | Henry Yin |
| National Institute of Neurological Disorders and Stroke | NS094754 | Henry Yin |

The funders had no role in study design, data collection and interpretation, or the decision to submit the work for publication.

## Author contributions
Konstantin I Bakhurin, Conceptualization, Formal analysis, Supervision, Investigation, Writing - original draft, Project administration, Writing - review and editing; Xiaoran Li, Conceptualization, Formal analysis, Supervision, Investigation, Writing - original draft, Writing - review and editing; Alexander D Friedman, Conceptualization, Investigation, Writing - original draft; Nicholas A Lusk, Formal analysis, Investigation; Glenn DR Watson, Namsoo Kim, Data curation, Formal analysis; Henry H Yin, Conceptualization, Formal analysis, Supervision, Writing - original draft, Project administration, Writing - review and editing

## Author ORCIDs
Konstantin I Bakhurin (ID) https://orcid.org/0000-0003-3660-4343
Henry H Yin (ID) https://orcid.org/0000-0003-1546-6850

## Ethics
Animal experimentation: Research was approved by Duke Institutional Animal Care and Use Committee (protocol A254-19-11).

## Decision letter and Author response
Decision letter https://doi.org/10.7554/eLife.54831.sa1
Author response https://doi.org/10.7554/eLife.54831.sa2

# Additional files

## Supplementary files
• Transparent reporting form

## Data availability
All data generated in this study are included in the manuscript.

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
