## [Decision Letter]

Thank you for submitting your article "Opponent regulation of action performance and timing by striatonigral and striatopallidal pathways" for consideration by *eLife*. Your article has been reviewed by three peer reviewers, and the evaluation has been overseen by a Reviewing Editor and Kate Wassum as the Senior Editor. The following individuals involved in review of your submission have agreed to reveal their identity: Ivan De Araujo (Reviewer #1); Hugo Merchant (Reviewer #2); David Kupferschmidt (Reviewer #3).

The reviewers have discussed the reviews with one another and the Reviewing Editor has drafted this decision to help you prepare a revised submission.

Summary:

The authors performed an exploration of the effects of optogenetic stimulation of direct and indirect pathway neurons in the ventrolateral striatum on licking behavior. The work reveals intriguing phenomena such as a "resetting" and "pausing" of the interval timing of licking behavior with direct and indirect pathway stimulation. The authors further show that direct pathway stimulation elicited licking while indirect pathway stimulation suppressed licking followed by a rebound licking post-stimulation. Furthermore, it was found that the direct pathway reorganized internal timing mechanisms associated with bouts of anticipatory licking, but direct indirect pathway stimulation transiently delayed the initiation of upcoming bouts of licking. The authors conclude that direct and indirect basal ganglia pathways exert opposing influences on internal timing mechanisms linked to goal-directed actions.

Essential revisions:

1) Were the optogenetic trials reported in the figures the first (and only) ones after animals had learned the task. If not, did the changes in timing attenuate upon performing the optogenetic task repeatedly?

2) Data shown in Figures 2J and especially 5J,K are surprising (i.e. the relative lack of effect of light on licking during consumption). Is it possible that the optical neuronal activation introduced estimation errors into the timing processes (which were then ignored when the actual taste was delivered), instead of directly acting upon lick pattern generators?

3) It is crucial to measure the width, not only the peak, of the lick rate distribution after the direct stimulation in the probe trials. A hallmark of interval timing is the scalar property, measured as an increase in the width of the licking responses as a function of timed interval. Hence, changes in the width of the response distribution could tell us whether the direct/indirect pathways affect this timing property. Is the stimulation of the direct and indirect pathways changing the skewness of the responses?

4) An analysis of phase locking between the laser pulses and licking needs to be included along with a detailed report of the onset latency of licking following each laser pulse at different stimulation rates. Furthermore, it appears that the licking rebound can occur before the predictive probe licking distribution. A two-mean Gaussian mixture model will determine the two modes (rebound and main probe licking distributions), by providing their distance and widths.

5) Please discuss the possibility that an opponent regulation of timing by the direct and indirect pathways can occur.

6) Indirect pathway stimulation produces a suppression of anticipatory licking (Figure 5C,G) but only decreases consummatory licking. What is the precise role of this pathway on the initiation and execution of licking?

7) The duty cycle hypothesis could be strengthened by an additional experiment where licking is measured as a function of the duration of the direct/indirect stimulation.

8) It has been suggested that time is encoded in the dynamic recruiting of cell populations in the corticostriatal circuit and that these patterns of activation scale with the duration of the timed interval (Crowe et al., 2014; Mello et al., 2015; Wang et al., 2018). How do the present results fit with this notion?

9) How can the authors rule out that the effects of stimulation change as a function of multiple prior bouts of light stimulation within a session? along the same lines, how many days was the optogenetic stimulation delivered to each mouse?

10) Ventrolateral striatum and direct/indirect pathway spiny projection neurons have long been implicated in behavioral reinforcement. How is this concept involved in the results?

11) Please include statistical analyses where pertinent. For example, the Results state that the authors were "less able to suppress licking that occurred following reward", but in this case and others, the effects of different time points of stimulation (e.g. prior to and during reward) were never directly compared.

---

## [Author Response]

Essential revisions:1) Were the optogenetic trials reported in the figures the first (and only) ones after animals had learned the task. If not, did the changes in timing attenuate upon performing the optogenetic task repeatedly?

The study contained 8 animals in each experimental cohort (D1-cre and A2A-cre), in addition to 5 control animals. Mice received all combinations of stimulation parameters, meaning that animals received at least 12 (4 stim frequencies x 3 trial timepoints) days of optogenetic manipulation, in addition to additional experiments performed to evaluate the effects of stimulation on timing. Stimulation parameters were chosen randomly for each animal on each day. As shown in Author response image 1, animals could receive stimulation multiple days in a row without significant detriment to their timing performance. The figures show peak trials from 4 consecutive calendar days in 4 example animals (2 A2A-cre mice, 2 D1-cre mice). To quantify intact timing performance over days, included are summary analyses on the peak trials that did not receive laser stimulation. Shown are the mean peak time, mean peak width, and the fraction of trials with a valid peak (peak with a minimum licking rate of 3 Hz and a minimum width of 2 seconds), demonstrating that timing performance was not affected by repeated exposure to light stimulation in either group.

We have expanded our description of these procedures in the Materials and methods section to emphasize that animals received repeated laser stimulation sessions:

“Mice received multiple sessions with stimulation in order to relate the effects of stimulation frequency and timing relative to reward on licking, meaning that animals received at least 12 (3 times in the trial x 4 stimulation frequencies) sessions of stimulation, in addition to additional tests to investigate the effects on timing. Stimulation parameters (stimulation frequency and timing relative to reward) were consistent within a session, but the order of stimulation was semi-randomized between mice. Stimulation could be repeated over multiple consecutive sessions without detriment to timing (data not shown).”

2) Data shown in Figures 2J and especially 5J,K are surprising (i.e. the relative lack of effect of light on licking during consumption). Is it possible that the optical neuronal activation introduced estimation errors into the timing processes (which were then ignored when the actual taste was delivered), instead of directly acting upon lick pattern generators?

While we had not considered this interpretation of our results, several lines of evidence argue in against the idea that stimulation results in a timing error. First, we showed that there was a strong relationship between the optical stimulation frequency of the direct and indirect pathways and the degree to which licking was increased/decreased. The timing interpretation would need to incorporate an explanation for how timing and licking rate are related. To our knowledge, such a relationship has not been described previously.

Second, we found a significant effects of direct pathway stimulation as a function of stimulation time on licking rate change (new Figure 2M is reproduced in our response to concern #11). The relatively small impact on licking during consumption is related to a ceiling effect on the licking pattern generators – during consumption, animals are near their maximum licking frequency (~ 10 Hz). We can only increase this rate slightly above the natural consumption rate.

Third, in our analyses performed in the frequency domain, we observed that stimulation of the direct pathway at could regulate the duty cycle of the on-off time of the licking pattern generators. Whereas licking frequency could not be raised much higher beyond that seen during consumption, we found that the power of the stimulation-evoked licking frequency was sensitive to stimulation frequency, even after reward. This is the first time this effect has been reported and strongly supports our interpretation that stimulation of the basal ganglia pathways directly interacts with the licking pattern generators. Our results provide evidence for an integration mechanism that couples basal ganglia output with the licking pattern generators in the brainstem. We have included an additional figure to explain this proposed mechanism, and have presented in the Discussion the connection between the present findings and our previous work on position control using integration in basal ganglia output nuclei.

In addition, we have expanded our Discussion to emphasize that basal ganglia activity *also* mediate interval timing. Their operation contributes to both behavioral output and timing processes.

3) It is crucial to measure the width, not only the peak, of the lick rate distribution after the direct stimulation in the probe trials. A hallmark of interval timing is the scalar property, measured as an increase in the width of the licking responses as a function of timed interval. Hence, changes in the width of the response distribution could tell us whether the direct/indirect pathways affect this timing property. Is the stimulation of the direct and indirect pathways changing the skewness of the responses?

We have performed the suggested analyses for both the direct and indirect pathway stimulation experiments. We found that direct pathway stimulation reduced the width of the peak (two-way ANOVA, main effect of stimulation, *F* = 20.05, p < 0.0001). However, we found that this effect was uniform as a function of time of stimulation (one-way, repeated measures ANOVA, *F* = 1.5, p = 0.28). This figure panel has been incorporated into Figure 9C. We note that peak width change during indirect pathway stimulation was already included in Figure 10F.

We also measured the skewness of the peaks and found that stimulation of neither the direct nor indirect pathways had significant impact on the distributions’ skew. These panels have been included in Figures 9D and 10G.

4) An analysis of phase locking between the laser pulses and licking needs to be included along with a detailed report of the onset latency of licking following each laser pulse at different stimulation rates. Furthermore, it appears that the licking rebound can occur before the predictive probe licking distribution. A two-mean Gaussian mixture model will determine the two modes (rebound and main probe licking distributions), by providing their distance and widths.

We have performed phase-locking analysis and latency analyses for 10 Hz direct pathway stimulation. We found that phase-locking did occur, but that the general phase of entrainment for each animal could vary across the population. For example, in Author response image 2, we provide two representative polar plots and licking rasters to indicate how for different animals, the interaction between laser pulses and licking could be offset – one shows strong locking at approximately 150 degrees, whereas another animal shows it at 315 degrees. We summarize the findings in a dot plot which reveals that this was true for 10Hz stimulation at different time points in the animals. To us, this reinforces the idea that direct pathway activity may regulate the operation frequency of the lower level licking CPGs by increasing its activation above a threshold via an integrator, but that BG output does not directly determine the cycle pattern for each individual lick. We also analyzed the latency to each lick following each laser pulse at 10 Hz stimulation. A two-way, mixed ANOVA found a significant effect of pulse number, but no main effect of time of stimulation.

We have decided to not include this data as an additional figure, as it does not add significantly to our main findings: 1) laser stimulation of the direct pathway boosts the licking frequency to a higher state, and 2) higher stimulation frequencies result in a greater proportion of licking to occur at the higher state, which is reflected in the power of the licking oscillation. This was already communicated effectively in Figure 3.

**Author response image 2. respfig2:** 

We add that laser stimulation at 5 Hz did not alter phase lock licking, as we did not see a reduction of licking frequency to 5 Hz. Furthermore, performing phase locking or latency analysis following each laser pulse on stimulations that exceed 10Hz was not possible with our dataset, as licking frequency could not exceed 10 Hz. For these reasons it cannot be assumed that a given lick that follows a laser pulse immediately preceding it was caused by the pulse.

5) Please discuss the possibility that an opponent regulation of timing by the direct and indirect pathways can occur.

We have added additional discussion of opponent regulation of timing in the Discussion, emphasizing that the two pathways seem to differentially regulate the timing mechanism, with the direct pathway acting to reset the clock and the indirect pathway to pause it. This relationship to timing processes may be a direct consequence of the presence of an integrator in the basal ganglia output. We have included more discussion of the traditional accumulator-based models for interval timing and how the presence of an integrator is compatible with those models.

6) Indirect pathway stimulation produces a suppression of anticipatory licking (Figure 5C,G) but only decreases consummatory licking. What is the precise role of this pathway on the initiation and execution of licking?

We have outlined the precise role of the indirect pathway in the execution of licking behavior in paragraphs four and five of the Discussion. Balanced activity of the direct and indirect pathways activity may keep the output of the system within a specific range. Our artificial stimulation of the indirect pathway creates a strong imbalance in the system, resulting in the suppression of the behavior.

“Artificially driving the direct pathway efficiently fills the integrator, and this results in near saturation even at relatively low stimulation frequencies, leading to animals achieving near-maximum licking rates. […] The balanced relationship between these two pathways may be responsible for the continuous adjustments to specific kinematic details of ongoing movement.”

7) The duty cycle hypothesis could be strengthened by an additional experiment where licking is measured as a function of the duration of the direct/indirect stimulation.

Our finding is that increasing stimulation frequency regulates the power of the licking oscillators’ cycle, the internal operation of which determines the duty cycle. This demonstrates how a wide range of activity in the basal ganglia (0-50Hz firing rate) can regulate the operation of a system that cycles at a much lower maximum rate (maximum cycle of licking centers: 10Hz).

We have made a stronger effort to explain that direct pathway activity is not directly responsible for itself regulating the duty cycle of the licking CPGs. We have included a new Figure 11 to convey the concept of integration in the basal ganglia output nuclei and how they may influence the operation of the lower level licking generation centers and have discussed this model thoroughly in the Discussion (paragraphs three and four). The integrator’s fill level is reflected in the on-time of the CPGs and potentially the cycle frequency as well up to a natural physical limit of 10 Hz.

“A variety of evidence suggests that the output nuclei of the BG may function as an integrator (Barter et al., 2015, Yin, 2014, Yin, 2017). […]However, mice rarely enter this regime on their own, perhaps due to the concurrent activity of the indirect pathway.”

8) It has been suggested that time is encoded in the dynamic recruiting of cell populations in the corticostriatal circuit and that these patterns of activation scale with the duration of the timed interval (Crowe et al., 2014; Mello et al., 2015; Wang et al., 2018). How do the present results fit with this notion?

We have added discussion of how our results may relate to these models for timing, but introduce what we feel are major limitations of these models in explaining timing behavior. Many implementations of the dynamic population-clock-based models fail to incorporate the known anatomical connections and physiological functions of the basal ganglia pathways into their operation.

“Some have suggested that the brain’s ability to time intervals is an emergent property of sequential neural population dynamics (Bakhurin et al., 2017, Crowe et al., 2014, Mauk and Buonomano, 2004, Mello et al., 2015). However, these models do not explain why specific neural circuits are critical for timing; nor do they incorporate detailed anatomical organization such as the direct and indirect pathways, which are shown to have distinct roles in timing. Our observations support classic pacemaker-accumulator models (Gibbon et al., 1984), which contain specific mechanisms that can be paused or reset (Buhusi and Meck, 2002). In fact, the proposed BG integrator mechanism is quite similar to the pacemaker-accumulator model, which also makes use of an integrator.”

“The presence of distinct mechanisms for resetting and pausing an internal clock could be related to the interaction of the direct and indirect pathways to regulate the integrator in the BG output nuclei. The opponent impact that direct and indirect pathways have on licking is due to their opponent roles in filling and emptying the integrator.”

“Some have argued that the accumulator is distributed in the cortex (Wang et al., 2018). Our results, however, suggest that it resides in the BG output circuit, though the underlying mechanisms for integration remain unclear. Future work will be necessary to elucidate the neural basis of this integrator and how the BG output influence and modulate targets in the midbrain and thalamus to regulate performance and timing of actions.”

9) How can the authors rule out that the effects of stimulation change as a function of multiple prior bouts of light stimulation within a session? along the same lines, how many days was the optogenetic stimulation delivered to each mouse?

Over time, laser stimulation has a reduced efficacy in evoking licking (via direct pathway stimulation) or rebound licking (following offset of indirect pathway stimulation). The possibility that these effects are due to repeated laser stimulation can be ruled out by observing that higher stimulation frequencies are capable of overriding these motivational effects. It is only at lower stimulation frequencies where we can observe the reduced impact that stimulation has on behavior. If the repeated delivery of light could lead to a reduced efficacy over time, the higher stimulation frequencies would have displayed this effect too. Instead the results are more consistent with the interpretation that basal ganglia pathways can modulate brain stem CPGs above an activation threshold. With time, as animals become more satiated, the threshold to activate the CPG is increased, and can only be overcome with higher-frequency stimulation.

We have expanded the relevant section in the Discussion (paragraph six):

“The effects of direct and indirect pathway stimulation were sensitive to motivational state, as the latencies to licking, overall licking rates, and the latency to rebound all changed over the course of the session (Figure 7). This could not be explained as simply a reduction in the efficacy of laser stimulation over the course of a single session because stimulation of the direct pathway at higher frequencies could override the effects of reducing motivation. These results can also be explained by the possibility that, over time, the activation threshold for licking increases.”

10) Ventrolateral striatum and direct/indirect pathway spiny projection neurons have long been implicated in behavioral reinforcement. How is this concept involved in the results?

It is critical to point out that the region that we describe as ventrolateral striatum (VLS) is anatomically distinct from the ventral striatum. Whereas the accumbens receives projections from the frontal cortical areas and limbic structures such as the amygdala and hippocampus, the VLS receives projections from sensory and motor cortex, and its projections target the substantia nigra pars reticulata, particularly the lateral part.

To our knowledge, the role of the VLS in reinforcement has not been explicitly demonstrated. Several studies have shown that stimulation of the direct pathway is reinforcing of a specific operant behavior, whereas indirect pathway stimulation is aversive, or reduces the likelihood of the operant response. These studies were conducted in the dorsomedial striatum (Kravitz et al., 2012) and in the dorsal striatum (Vicente et al., 2016). Because of the location differences of manipulation, it is difficult to draw specific claims about how the orofacial zone would follow these patterns from our data. This is in addition to the lack of any kind of operant behavior in our task that would trigger the stimulation, since stimulation was delivered at specific time points, not contingent upon the animals’ behavior.

A number of observations are inconsistent with the idea that our manipulations are driving reinforcement-related signals. First, as addressed in our response to comment 1, we did not see any significant changes that accumulated over time as a result of repeated stimulations of either pathway. Second, as we describe in our answer to comment 2, the frequency of direct pathway activity is related to the amount of change in licking, and that licking can be time-locked to stimulation of the direct pathway at 10 Hz, suggesting a direct role of the direct pathway in interacting with the licking generating, motor system.

11) Please include statistical analyses where pertinent. For example, the Results state that the authors were "less able to suppress licking that occurred following reward", but in this case and others, the effects of different time points of stimulation (e.g. prior to and during reward) were never directly compared.

This observation did not escape our notice and we have emphasized this observation more in the manuscripts’ results (paragraphs two and five) and Discussion sections. The relationships between the direct and indirect pathways and the initiation and execution of licking behavior is addressed in paragraphs seven and eight of the Discussion. In accord with the helpful suggestion by the reviewers, we have included the results of additional analyses to support our statements about the relative effects of stimulation at different time points. For stimulation of both the direct and indirect pathways, we observed significant effects of stimulation time on the change in lick rate. We have reported the results of the ANOVA analysis performed to compare the effects of varying stimulation time in the Results section. See also Figure 2M and Figure 5M.